

# What's streamflow got to do with it? A probabilistic simulation of the competing oceanographic and fluvial processes driving extreme along-river water levels

Katherine A. Serafin[1,3], Peter Ruggiero[1], Kai A. Parker[2], and David F. Hill[2]

[1]College of Earth, Ocean, and Atmospheric Sciences, Oregon State University, Corvallis, OR, USA
[2]College of Engineering, Oregon State University, Corvallis, OR, USA
[3]Present Address: Department of Geophysics, Stanford University, Stanford, CA, USA

**Correspondence:** Katherine A. Serafin (kserafin@stanford.edu)

**Abstract.** Extreme water levels driving flooding in estuarine and coastal environments are often compound events, generated by many individual processes like waves, storm surge, streamflow, and tides. Despite this, extreme water levels are typically modeled in isolated open coast or estuarine environments, potentially mischaracterizing the true risk to flooding facing coastal communities. We explore the variability of extreme water levels near the tribal community of La Push, within the Quileute

Indian Reservation on the Washington state coast where a river signal is apparent in tide gauge measurements during high discharge events. To estimate the influence of multivariate forcing on high water levels, we first develop a methodology for statistically simulating discharge and river-influenced water levels in the tide gauge. Next, we merge probabilistic simulations of joint still water level and discharge occurrences with a hydraulic model that simulates along-river water levels. This methodology produces water levels from thousands of combinations of events not necessarily captured in the observational record.

We show that the 100-yr ocean or 100-yr streamflow event does not always produce the 100-yr along-river water level. Along specific sections of river, both still water level and streamflow are necessary for producing the 100-yr water level. Understanding the relative forcing of extreme water levels along an ocean-to-river gradient will better prepare communities within inlets and estuaries for the compounding impacts of various environmental forcing, especially when a combination of extreme or non-extreme forcing can result in an extreme event with significant impacts.

## 15    1    Introduction

Storm events often generate concurrently large waves, heavy precipitation driving increased streamflow, and high storm surges, making the relative contribution of the actual drivers of extreme water levels difficult to interpret from tide gauge observations alone. Studies at the global (e.g., Ward et al. (2018)), national (e.g., Wahl et al. (2015); Svensson and Jones (2002); Zheng et al. (2013)) and regional scale (e.g., Odigie and Warrick (2017); Moftakhari et al. (2017)) have evaluated the likelihood for

variables such as high river flow and precipitation to occur during high coastal water levels, demonstrating that relationships often exist between these individual processes. Understanding the nature of the dependency between bivariate or multivariate processes is one of the first steps in piecing together the contributors to flooding events.




Around river mouths, the elevation of the water level measured by tide gauges, or the still water level (SWL), varies depending on the mean sea level, tidal stage and the non-tidal residual contributors which may include the following forcings; storm surge, seasonally-induced thermal expansion (Tsimplis and Woodworth, 1994), the geostrophic effects of currents (Chelton and Enfield, 1986), wave setup (Sweet et al., 2015; Vetter et al., 2010), and river discharge. Most commonly, estimates of
non-tidal residuals are determined by subtracting predicted tides from the measured water levels. However, residuals computed in this way often contain artifacts of the subtraction process from phase shifts in the tidal signal and/or timing errors (Horsburgh and Wilson, 2007). Another approach to describing the non-tidal residual is the skew surge, which is the absolute difference between the maximum observed water level and the predicted tidal high water (de Vries et al., 1995; Williams et al., 2016; Mawdsley and Haigh, 2016). While this methodology removes the influence of tide-surge interaction from the non-tidal
residual magnitude, it does not differentiate between the many factors contributing to the water level, an important step for distinguishing when and why high water (and thus flooding) is likely to occur.

Hydrodynamic models have recently been used in attempts to quantify the relative importance of river and ocean-forced water levels to flooding. The nonlinear coupling of wind and pressure driven storm surge, tides, wave-driven setup, and riverine flows has been found to be a vital contributor to overall water level elevation (Bunya et al., 2010). Furthermore, river discharge
is often found to interact nonlinearly with storm surge (Bilskie and Hagen, 2018), exacerbating the impacts of coastal flooding (Olbert et al., 2017), which suggests that the extent or magnitude of flooding is often underpredicted when both river and oceanic processes are not modeled (Bilskie and Hagen, 2018; Kumbier et al., 2018; Chen and Liu, 2014). The computational demand of two and three-dimensional hydrodynamic models, however, typically precludes a large amount of events to be examined. Therefore, while accurately modeling the physics of the combined forcings, researchers taking this approach are
often limited to modeling only a few select cases.

This study explores the influence of oceanographic and fluvial processes driving extreme water levels along a coastal river where there is a substantial fluvial signal recorded in the tide gauge. Our study site, the Quillayute River, terminates in the Pacific Ocean at La Push, Washington, an incorporated tribal community within the Quileute Indian Reservation. In order to better understand the river- and ocean-forced water levels at this location, a methodology is developed for defining and
removing river-influenced water levels from SWLs measured at tide gauges. Both river discharge and river-influenced water levels are then incorporated into a non-stationary, probabilistic total water level model. This allows for multiple synthetic representations of joint ocean and fluvial processes that may not have occurred in the relatively short observational records. Next, a 1-dimensional hydraulic model is used to simulate water surface elevations along a 10 km stretch of river. Surrogate models are generated from the hydraulic model simulations and used to extract along-river water levels for each probabilistic
joint-occurrence of SWL and river discharge. Finally, spatially-varying extreme event return levels are derived and discussed. The following sections describe the study area, present the modeling framework linking oceanographic and fluvial systems, and evaluate the compounding drivers of extreme water levels along this river system.



## 2   Study Area

The Quillayute River is located in Washington state along the US West coast and drains approximately 1630 km$^2$ of the northwestern Olympic Peninsula into the Pacific Ocean (Czuba et al., 2010). The Quillayute River is approximately 10 km long, is formed by the confluence of the Bogachiel and Sol Duc Rivers (Figure 1), and enters the Pacific Ocean at La Push, Washington, home to the Quileute Tribe. The Quileute Indian Reservation is approximately 4 km$^2$ and the majority of community infrastructure sits at the river mouth, bordering the river and open coast. The Quileute Harbor Marina is also situated just inside the river mouth, and is the only port between Neah Bay and Westport, Washington. Rialto spit, which connects Rialto Beach to Little James Island, contains a rocky revetment which protects the marina and the community from ocean and storm wave impact.

The Quillayute River is a natural, unstablized river that is relatively straight at the confluence of the Bogachiel and Sol Duc rivers and increases in sinuosity moving towards the river mouth. Channel-bed materials are coarse (gravel and cobble) in the free-flowing channels and dominated by sand in the small estuary (Czuba et al., 2010). Upstream of river km 3 there are numerous point bars and bends in the river. Between river km 1.5 and 3, the Quillayute is braided with several side channels, usually containing woody debris (Czuba et al., 2010). The channel is straight near the river mouth and is confined by the Rialto spit revetment before draining into the Pacific Ocean.

The oceanic climate of the coastal Pacific Northwest (PNW) is cool and wet with a small range in temperature variation and the majority of rainfall between October and May. Four river basins, the Sol Duc, Bogachiel, Calawah, and Dickey rivers, feed into the Quillayute River and comprise the majority of the watershed. Streamflow in the region is primarily from storm-derived rainfall in the winter and snowmelt during the spring and summer (WRCC, 2017).

Oceanographically-driven SWLs are generally comprised of non-tidal residuals, astronomical tides, and mean sea level. Regional variations in shelf bathymetry, shoreline orientation, storm tracks (Graham and Diaz, 2001), seasonality (Komar et al., 2011), and winds drive differences in storm surge along the US West coast. However, the narrow continental shelf, in relation to broad-shelved systems, controls the magnitude of storm surge, and it is rarely larger than 1 m (Bromirski et al., 2017; Allan et al., 2011). The PNW is also influenced by a unique interannual climate variability due to the El Niño Southern Oscillation. During El Niño years, the PNW experiences increased water levels for months at a time, along with changes in the frequency and intensity of storm systems (Komar et al., 2011; Allan and Komar, 2002). In the PNW, tides are micro- and mesotidal, and at La Push the tidal range is mixed, predominantly semidiurnal, with a mean range of 1.95 m and a great diurnal range of 2.58 m (https://tidesandcurrents.noaa.gov/datums.html?id=9442396).

Global rise in sea level and local changes in vertical land motions result in significant longshore variations of relative sea level along the Washington coastline. The northern Washington coast is experiencing relative sea level rates of -1.85 ± 0.42 mm/yr due to a rising coastline, while relative sea level in Willapa Bay in southern Washington is 0.94 ± 2.14 mm/yr (Komar et al., 2011). Tide gauge records at La Push are too short to calculate robust trends in sea level, however, sea level is likely rising in this location, rather than falling, partly due to local land subsidence (Miller et al., 2018).





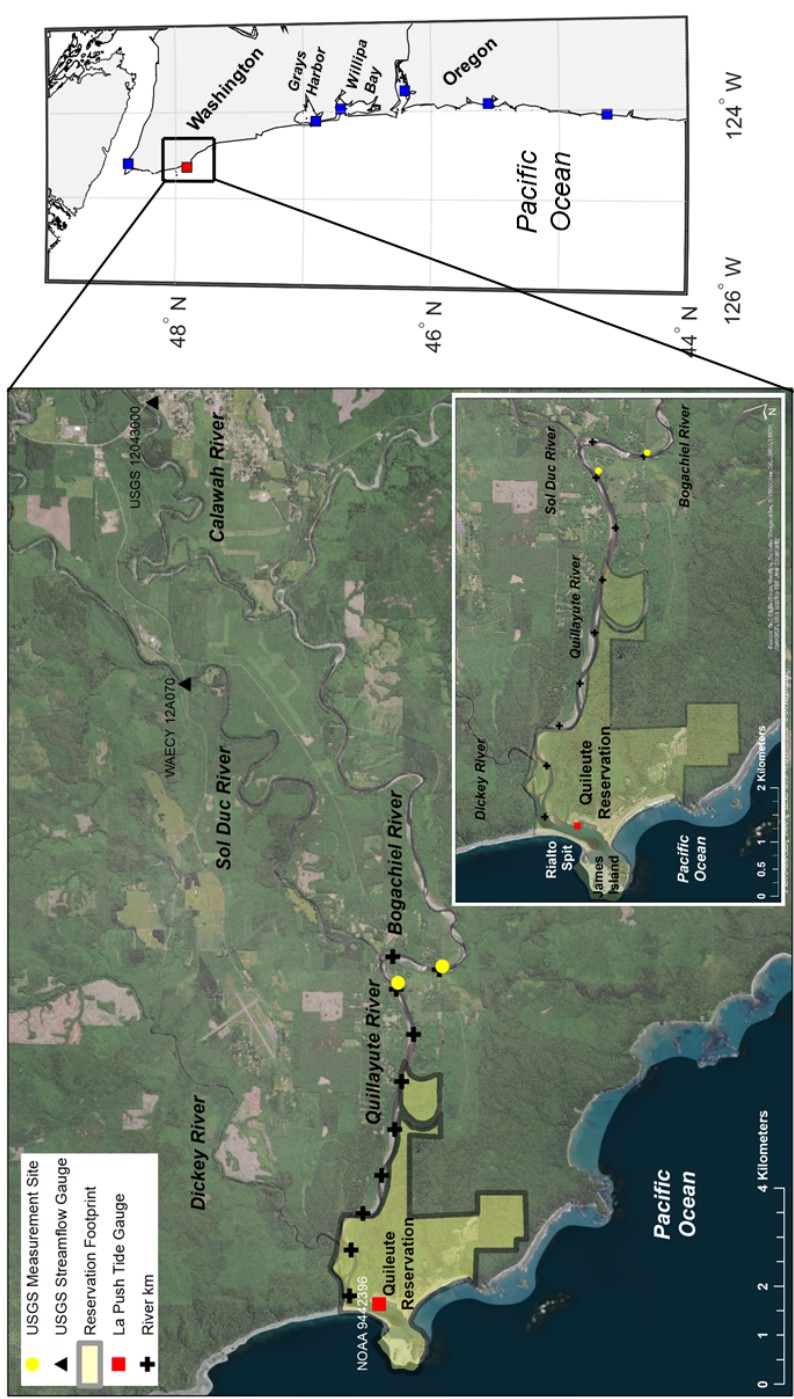

**Figure 1.** Map of study area (left), which is denoted on the regional map (right) in the black box. The La Push tide gauge is represented as a red square while other regional tide gauges are represented as blue squares. The Calawah and Sol Duc river gauges are represented as black triangles and USGS measurement sites from the May 2010 survey are depicted as yellow circles. Approximate river kilometers are denoted as black crosses on the study area map.



**Figure 2.** Digital Elevation Model (DEM) used for the HEC-RAS simulations of the Quillayute River. HEC-RAS cross sections are depicted as grey lines. Approximate river kilometer and the location of the tide gauge are depicted as diamonds and a square, respectively.





## 3   Data and methods

### 3.1   Modeling framework

Return level, or design, events are typically assessed via analyses of available observational datasets. However, observational records rarely extend more than a few decades, suggesting that all combinations of jointly-occurring processes generating

extreme water levels may not have been physically realized. Therefore, in order to understand the oceanographic and fluvial drivers of extreme water levels along the Quillayute River from a full range of possible forcing conditions, we develop a methodology to merge statistically simulated joint SWL and discharge records with a one-dimensional (1D) hydraulic river flow model. This approach is designed to allow for an interpretation of extreme water levels as if different physically plausible combinations of individual driving processes had been available to be sampled over the last thirty years.

First, a method is developed to define and model river-influence in the SWLs. Next, combinations of daily maximum SWL and river discharge are statistically simulated to create many random realizations of joint SWL-discharge forcing using the Serafin and Ruggiero (2014) full simulation total water level model. Care is taken to appropriately model both the non-stationarity of each signal, as well as the dependence between the signals. A range of SWL-discharge conditions are modeled using the US Army Corps of Engineers' (USACE) Hydrologic Engineering Center's River Analysis System (HEC-RAS; Brunner (2016)) to

produce surrogate models for generating along-river water levels. The surrogate models are then used to produce water levels at a series of transects for each statistically simulated SWL-discharge event. The synthetic SWL-discharge simulations paired with HEC-RAS water surface profiles allows for an analysis of the dominant drivers of extreme water levels along the river. Descriptions of the hydrodynamic and statistical models, as well as the overall framework for modeling spatially-varying water levels are described in the following sections.

### 3.2   Hydraulic model domain and setup

HEC-RAS is a model that is used to estimate water surface elevations in rivers and streams in both steady and unsteady flow and under subcritical, supercritical, and mixed flow regimes (Goodell, 2014). HEC-RAS has been previously used to model water surfaces for a range of applications including, but not limited to, floodplain mapping (Yang et al., 2006), flood forecasting (Saleh et al., 2017), dam breaching (Butt et al., 2013), and flood inundation (Horritt and Bates, 2002). HEC-RAS computes

water levels by solving the 1D energy equation with an iterative procedure, termed the step method, from one cross-section to the next (Brunner, 2016). For subcritical flows, the step procedure is carried out moving upstream; computations begin at the downstream boundary of the river and the water surface elevation at an upstream cross-section is iteratively estimated until a balanced water surface is obtained. Energy losses between cross-sections are comprised of a frictional loss via the Manning's Equation and a contraction/expansion loss via a coefficient multiplied by the change in velocity head (see Brunner (2016) for

more details).

In this application, HEC-RAS is used to model 1D water levels under gradually varied, steady flow conditions at specified transects along the Quillayute River. While a simplification of flood processes, this methodology is commonly used to create flood hazard maps. HEC-RAS model runs require detailed terrain information for the river network, including bathymetry and



topography for the floodplains of interest. Topography data is sourced from a 2014 U.S Army Corps of Engineers (USACE) lidar survey (USACE, 2014). Bathymetry data is developed by blending two NOAA digital elevation models (DEM): National Geophysical Data Center's (NGDC) La Push, WA tsunami DEM (1/3 arc second; NGDC (2007)) and the coastal relief model (3 arc seconds; NGDC (2003)). These datasets, however, do not accurately resolve the channel depths of the Quillayute River

inland of the coast, so a 2010 US Geological Survey (USGS)-conducted bathymetric survey of the river is also blended into the DEM (Czuba et al., 2010).

In 2010, depths of along-river cross sections and an 11 km long longitudinal profile from the Bogachiel River (Figure 1) to the mouth of the Quillayute River were surveyed (Czuba et al., 2010). The survey of the longitudinal river profile also recorded the elevation of the water surface. Ideally, the collected bathymetry dataset would be merged directly into the existing DEM.

The Quillayute River, however, is uncontrolled and meanders over time, producing a variation in the location of the main river channel between the DEM and the high-resolution USGS-collected bathymetric data. Therefore, the USGS bathymetric profiles are adjusted to match the location of the DEM channel. While a product of multiple datasets and processing steps, the final DEM provides bathymetric/topographic data with the most up-to-date channel depths for the Quillayute River (Figure 2).

A series of 58 transects are extracted from the DEM using HEC-GeoRas (Ackerman, 2009) and written into a geometric

data file for input into HEC-RAS (Figure 2). Each river transect extends across the floodplain to the 10 m contour, where applicable. Otherwise, each transect terminates at the highest point landward of the river. Because HEC-RAS computes energy loss at each transect via a frictional loss based on the Manning's equation, Manning's coefficients, an empirically derived coefficient representing resistance of flow through roughness and river sinuosity, are selected for the river channel and the floodbanks. In-channel Manning's coefficients are tuned to calibrate the model's resulting water surface elevations with that

of the observed water surface data (see section 3.2.1). Manning's coefficients for the rest of the computational domain (e.g., anything overbank) are estimated using 2011 Land Cover data from the Western Washington Land Cover Change Analysis project (NOAA, 2012) and visual inspection of aerial imagery. Model domain boundary conditions are chosen as the water surface elevation at the tide gauge (m; downstream boundary) and river discharge from a combination of records representing the Quillayute River watershed ($m^3s^{-1}$; upstream boundary).

**3.2.1    HEC-RAS model validation**

Observational records in the region are generally sparse; one tide gauge exists in the marina near the river mouth and hourly discharge measurements are only located on two of the four rivers which feed into the Quillayute watershed (Figure 1). The closest gauge is located 7 miles upriver from the Quillayute River on the Sol Duc River (WA Dept of Ecology 12A070) and measures approximately 9 years (2005-2014) of hourly discharge and stage observations. The second gauge is located on the

Calawah River (USGS 12043000), which flows into the Bogachiel River, and has hourly discharge and stage measurements from 1989 - 2016. While the Calawah River gauge is located approximately 15 miles upriver from the Quillayute River, the steep catchment drives a short response time in rainfall and the record is highly correlated with the discharge measurements from the Sol Duc River gauge.



In order to determine the dominant inputs to Quillayute River discharge, combined estimates of the Sol Duc and Calawah Rivers are compared to measurements taken on the Quillayute River in May 2010 (see Figure 1 for measurement location; Czuba et al. (2010)). Combined discharge estimates from the Sol Duc and Calawah rivers underpredict streamflow in the Quillayute River by approximately 33%. An area scaling watershed analysis (Gianfagna et al., 2015) is undertaken to rectify

the discharge by the amount of ungauged watershed. The watershed delineation shows that the Bogachiel, Calawah, Sol Duc, and Dickey rivers account for 24%, 22%, 37%, and 17% of the total Quillayute River watershed area. Noting the similar watershed characteristics and proportional area, the contribution of the Bogachiel River is estimated by scaling the Calawah River discharge measurements by a factor of 2.09. Combined discharge estimates from the Sol Duc River and Bogachiel River, computed using the above scaling factor, are also compared to the Quillayute discharge measurements taken during the 2010

survey. Using this methodology, the discharge estimates of the Quillayute River fall within the uncertainty of the discrete USGS measurements (Table 1).

The longitudinal measured water surface profile allows for the verification and calibration of HEC-RAS modeled water surface elevations on the day of the survey (Figure 3). HEC-RAS is run using discharge of the watershed-scaled Bogachiel River as the upstream boundary condition during the hour of the field survey and this discharge is combined with a lateral

inflow from the Sol Duc River around river km 8.5. Manning's coefficients along the Quillayute are calibrated to best represent the water surface elevation on the day of survey. The final calibrated HEC-RAS model produces a water surface elevation with an average bias less than 1% (less than 1 cm) and an average standard deviation of approximately 5% (7.5 cm). The maximum difference between the two water surfaces is approximately 14 cm (20%). The percent difference between the depth of the observed and modeled water surface is almost always less than 10% (Figure 3). Final Manning's coefficients range from

to 0.005 to 0.1, and are on average 0.025.

### 3.3   Total water level simulation model

Hourly measured SWLs and predicted tide measurements at the La Push tide gauge (NOAA station 9442396) relative to Mean Lower Low Water (MLLW) are downloaded, transformed into NAVD88 to match the DEM, and decomposed into mean sea level ($\eta_{MSL}$), tide ($\eta_A$), and non-tidal residual ($\eta_{NTR}$). The $\eta_{NTR}$ is further decomposed into monthly mean sea level

anomalies ($\eta_{MMSLA}$), seasonality ($\eta_{SE}$), and storm surge ($\eta_{SS}$), using methods described in Serafin et al. (2017). A 6th geophysical signal recorded by the tide gauge, the river-influenced water level ($\eta_{Ri}$), is also evaluated and removed from the $\eta_{SS}$ signal (see section 4.2 for description and methods).

The continuous La Push tide gauge record begins in 2004, recording 12 years of water levels. This record, however, does not capture the extreme water levels occurring during the 1982/83 and 1997/98 El Niños. Therefore, water levels from the La

Push tide gauge are merged with water levels from the Toke Point tide gauge (beginning in 1980, NOAA station 9440910) to create a combined water level record representing a larger range of extreme conditions. $\eta_A$ and $\eta_{SE}$, water level components deterministic to the La Push tide gauge, are extended to 1980. Water level components influenced by regional or local forcings like $\eta_{MMSLA}$ and $\eta_{SS}$, are compared before combining. $\eta_{MMSLA}$ between the Toke Point and La Push tide gauges are similar, so Toke Point $\eta_{MMSLA}$ are appended to the beginning of the La Push $\eta_{MMSLA}$. Toke Point, however, has slightly





**Table 1.** Quillayute River discharge measurements from the USGS survey (Czuba et al., 2010) compared to the Quillayute River discharge estimates computed by adding the Sol Duc USGS gauge measurements with the Bogachiel River discharge, estimated via scaling of the Calawah River gauge measurements. The parenthesis in the last column is the standard deviation of USGS survey measurements ($m^3 s^{-1}$).

| Date of Survey | Sol Duc ($m^3 s^{-1}$) | Calawah ($m^3 s^{-1}$) | Bogachiel ($m^3 s^{-1}$) (estimated) | Quillayute ($m^3 s^{-1}$) (estimated) | Quillayute ($m^3 s^{-1}$) (measured) |
|---|---|---|---|---|---|
| 4/20/2010 | 52 | 28 | 58 | 110 | 116 (7) |
| 4/21/2010a | 48 | 25 | 53 | 101 | 108 (1) |
| 4/21/2010b | 48 | 25 | 52 | 100 | 103 (3) |
| 4/21/2010c | 46 | 24 | 50 | 96 | 107 (1) |
| 5/4/2010a | 73 | 69 | 144 | 217 | 220 (5) |
| 5/4/2010b | 70 | 66 | 137 | 207 | 207 (4) |
| 5/5/2010 | 59 | 51 | 107 | 166 | 170 (3) |
| 5/6/2010 | 50 | 40 | 84 | 134 | 136 (3) |




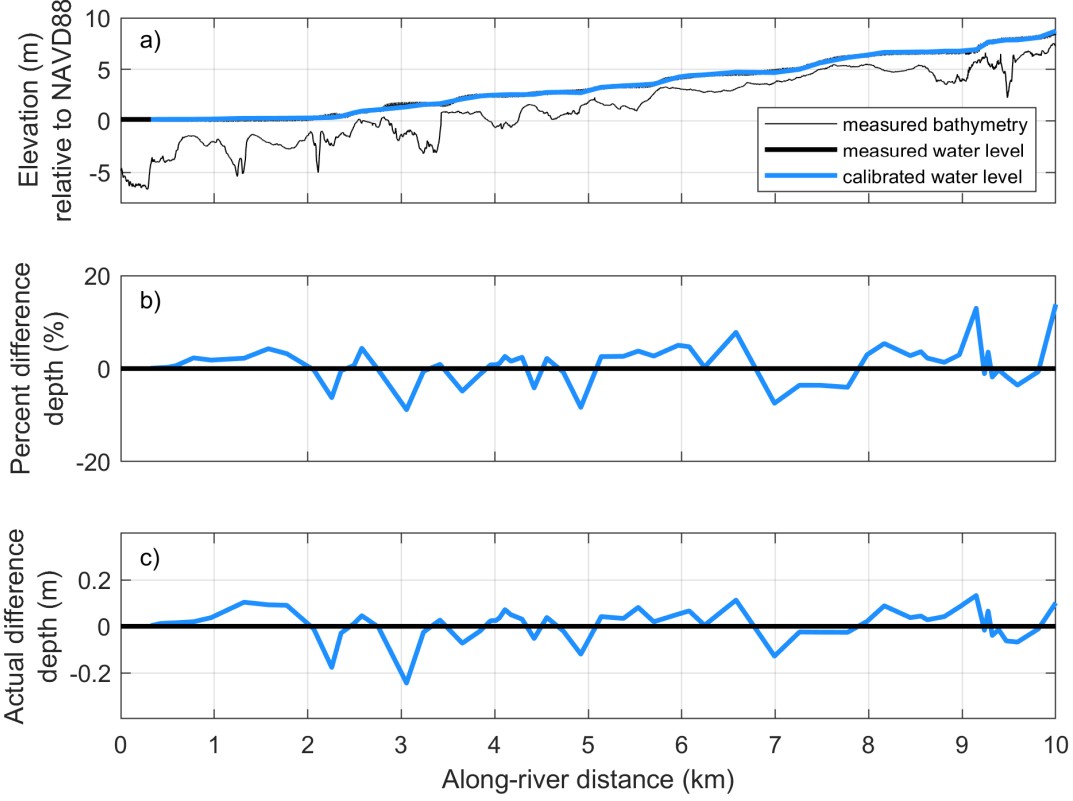

**Figure 3.** a) Bathymetry and longitudinal profile from the Bogachiel River to the mouth of the Quillayute River surveyed by the USGS in May of 2010 (black). The longitudinal water level for the calibrated HEC-RAS model is depicted in blue. b) Percent difference between the measured (black) and HEC-RAS modeled (blue) water level. c) Actual difference between the measured (black) and HEC-RAS modeled (blue) water level.

higher magnitude $\eta_{SS}$ than La Push and there is a noticeable offset in the highest $\eta_{SS}$ peaks. A correction is thus applied to the Toke Point $\eta_{SS}$ before appending it to the beginning of the La Push $\eta_{SS}$. $\eta_{MSL}$ is extended back to 1980 using relative sea level rise trends for the region. Once the two tide gauges are merged, the combined hourly tide gauge record extends from 1980 - 2016 and is 97% complete. Discharge measurements sampled at 15 minute intervals for the Calawah and Sol Duc rivers

5    are interpolated to hourly increments to match the timing of the SWL measurements. At the hourly scale, the Calawah River record is 99% complete, while the Sol Duc River record is 100% complete.

The non-stationary, probabilistic full simulation model of Serafin and Ruggiero (2014) (hereinafter, SR14) was developed to produce synthetic time series of total water levels (TWLs), the combination of waves, tides, and non-tidal residuals, on open-coast sandy beaches. SR14 simulates the individual components of the TWL in a Monte Carlo sense, while appropriately

10   accounting for any dependencies existing between the variables. This modeling technique is able to include non-stationary





processes influencing extreme and non-extreme events, such as seasonality, climate variability, and trends in wave heights and water levels. SR14 outputs a number of synthetic records of all variables driving TWLs that produce alternate, but physically plausible, combinations of waves and water levels along an identified stretch of coastline (see SR14 and Serafin et al. (2017) for more information). This technique is flexible to allow for both the simulation of the present-day climate for computing

robust statistics on extreme TWL events, as well as the simulation of future climates and their impact on extreme TWLs.

Because SR14 was developed for use in open-coast environments, it does not include a procedure for simulating estimates of river discharge, important to high water levels in estuarine environments, as well as present in the local tide gauge at the La Push study site. SR14 is therefore modified to produce synthetic time series of discharge as well as a river-induced water level. Specifics of these modifications are presented in section 4.3.

### 3.4   Hybrid modeling of along-river water levels

The modified simulation technique of SR14 is used to produce 70 500 year long synthetic records representing present-day climate for the time periosd of 1980-2016 of daily maximum SWL and discharge for both the Sol Duc and Bogachiel rivers. Modeling all of the simulated conditions in HEC-RAS in order to output along-river water levels would be prohibitively expensive. As an alternative to time consuming simulations, surrogate models (Razavi et al., 2012) are developed to approximate

the response of a HEC-RAS simulation. A large number of combinations of SWL and river discharge at the Bogachiel and Sol Duc rivers are run in HEC-RAS, outputting along-river water level at each HEC-RAS transect. The number of combinations of SWL and river discharge used in the surrogate models are chosen to minimize interpolation errors during validation runs. A surrogate model representing along-river water level is created for each modeled SWL condition using a scattered linear interpolation of the 3D surface of boundary conditions.

Along-river water levels are extracted from the surrogate model relating to each synthetic combination of SWL and river discharge, providing a longitudinal water surface profile for each day of the 500 year long record in an efficient manner. The large sample size of joint SWL-discharge events ensures a robust, probabilistic estimate of low probability water levels along the Quillayute River. This allows for an exploration of the drivers of along-river water levels over the past 35 years.

### 3.5   Extracting spatially variable return level events

The new methodology described in this paper allows for a statistically robust estimate of low probability, along-river water levels not observed in the historical record. Typically, return levels are estimated by modeling the estimated 100-yr hydrologic or meteorologic event, and the resulting water level is assumed to be statistically representative of this condition. However, processes driven by multiple variables means that different "sizes" of hydrologic conditions could potentially drive low probability water levels. The 500 year long synthetic records simulated using the modified SR14 allows for the empirical extraction

of return level events rather than an estimation from historic records. Using the count-back method, SWL, river discharge, and water level return level events are selected from each record, where the largest, 5th largest, and 10th largest events in each record correspond to the 500-yr, 100-yr, and 50-yr return levels, respectively, at each transect. This allows for an analysis of spatially-variable, along-river extreme water levels, as well as the ability to map to the jointly-occurring forcings driving the




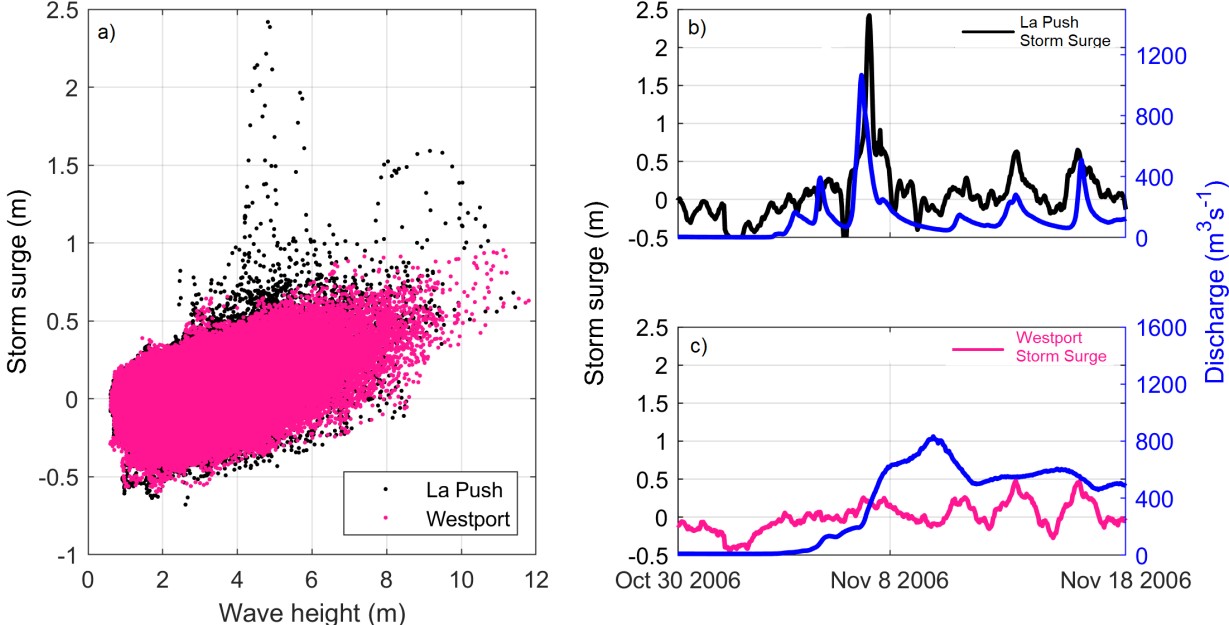

**Figure 4.** a) The joint relationship between storm surge and wave height for La Push, Washington (black) and Westport, Washington (pink). Example storm surge and discharge relationship at b) La Push and c) Westport, Washington.

return level water surface. The large sample space of simulated variables permits a comparison of event-based return levels, where the 100-yr water level is determined by the 100-yr forcing, to response-based return levels, where the 100-yr water level is derived.

## 4 River-influence in the tide gauge

5  Once the observational SWL at the La Push tide gauge is decomposed, peak $\eta_{SS}$ events are found to be the highest on record compared to all US West coast tide gauge stations (Serafin et al., 2017). $\eta_{SS}$ is often found to be jointly related to significant wave height (Hs), where the most extreme $\eta_{SS}$ occur during storms with associated low pressures, high winds, and high waves. When compared to the relationship of Hs and $\eta_{SS}$ towards the south in Westport, Washington, many large $\eta_{SS}$ at La Push occur during small waves, outside of the joint Hs-$\eta_{SS}$ relationship (Figure 4).

10  Upon further investigation of the La Push $\eta_{SS}$ record, almost all instances of extreme $\eta_{SS}$ events irregular to the joint Hs-$\eta_{SS}$ relationship are positively correlated with high discharge events. This is inconsistent with $\eta_{SS}$ in Westport, Washington (Figure 4) and with other tide gauges along the US West coast (not shown). Most tide gauges in Washington and Oregon are situated in bays and estuaries where the estuary volume is much larger than the river input volume. On the other hand, the La





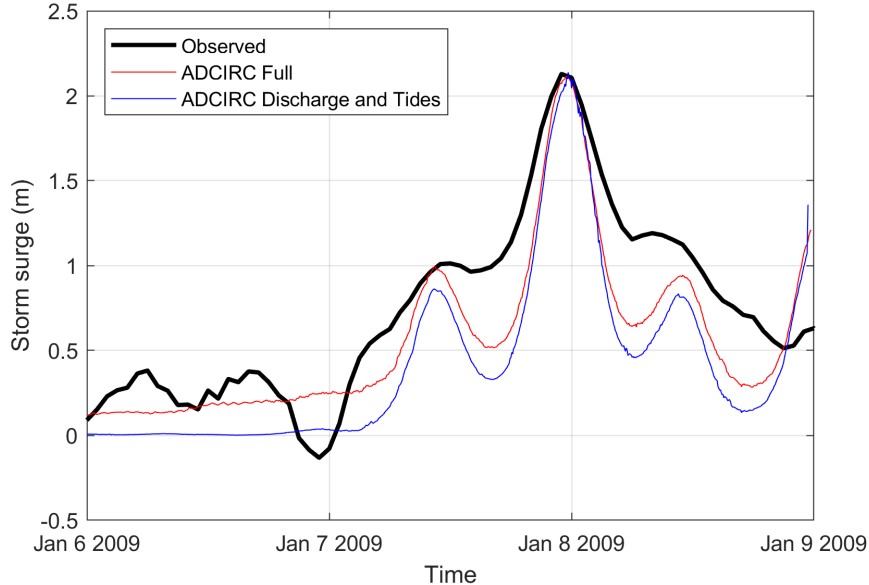

**Figure 5.** Resulting storm surge at the La Push tide gauge modeled using ADCIRC for a simulation including full forcing (red) and a simulation including only discharge and tides (blue) compared to the observed storm surge (black). The ADCIRC simulation was run for the maximum discharge event on record occurring on January 8, 2009.

Push tide gauge is located on a river discharging directly into the ocean. It is therefore hypothesized that the anomalously large signal in the $\eta_{SS}$ is indeed river-induced.

## 4.1 Physics-based evidence of river-induced signal

To further investigate the anomalously large $\eta_{SS}$ at the La Push tide gauge, the hydrodynamic model ADvanced CIRCculation
(ADCIRC, Luettich Jr et al. (1992)) and Simulating Waves nearshore (SWAN, Zijlema (2010) model (ADCSWAN; Dietrich et al. (2011)) is used to simulate an example storm event. ADCSWAN has been extensively validated worldwide and has recently found to be skillful for modeling $\eta_{SS}$ in the PNW (Cheng et al., 2014). ADCIRC is run in 2D depth-integrated barotropic mode which performs well for calculating water surface elevations during storm events (Weaver and Luettich, 2010). SWAN is run in non-stationary mode on an unstructured grid, allowing for tight coupling to ADCIRC.
To test the influence of streamflow on water levels at the tide gauge, the peak streamflow event on record, occurring on January 8, 2009, is simulated. The model is run with two forcing implementations: one including full forcing (e.g., waves, wind, pressure, streamflow, sea level anomalies, seasonality, and tides) and one including only streamflow and tides. Model results show that the simulation including only streamflow and tides is nearly able to recreate the measured $\eta_{SS}$ signal at the tide gauge (Figure 5). The addition of wind, pressure, waves, sea level anomalies, and seasonality is found to have minimal
impact on the peak observed $\eta_{SS}$. Furthermore, maximum peak $\eta_{SS}$ is found to occur during low tide, indicating a relationship





between tide and discharge. While this simulation only explores one instance of this phenomenon, it provides physics-based evidence that anomalously high $\eta_{SS}$ at this tide gauge is likely being driven by large discharge events.

## 4.2 Removal of river-influence from the oceanographic signal

Storms tend to influence large stretches of coastline at once, and while site-specific variations in the coastline or distance
from storm can drive local variations in the amplitude of $\eta_{SS}$, the overall $\eta_{SS}$ signal is fairly coherent across regional tide gauges across the PNW. The river-influenced water levels are therefore isolated and removed from the La Push $\eta_{SS}$ record by developing a relationship between the La Push $\eta_{SS}$ and a regionally-averaged $\eta_{SS}$.

$\eta_{SS}$ decomposed from the Neah Bay, Westport, Astoria, South Beach, and Garibaldi tide gauges are averaged each hour to create a regional $\eta_{SS}$ record (black line; Figure 6; tide gauge locations in Figure 1). The standard deviation ($\sigma$) of the available
$\eta_{SS}$ records at each hour is used to represent the variability of $\eta_{SS}$ due to local effects at each station. $\eta_{SS}$ at La Push that are larger than the regional average $+ 2.5\sigma$ are considered anomalous to the region, and defined as river-influenced water levels ($\eta_{Ri}$). Observations flagged as larger than the regional average $+ 2.5\sigma$ (dashed line; Figure 6) were replaced with the regional average $+ \sigma$. A value of $+ \sigma$ was chosen to minimize jumps in time series when subsituting in a smoother dataset. While this methodology does not remove all the effects of $\eta_{Ri}$ in the $\eta_{SS}$ signal, it captures the majority of anomalous water levels driven
by high discharge events.

$\eta_{Ri}$ is produced from the difference between the original La Push $\eta_{SS}$ and the $\eta_{SS}$ modified described above which removes $\eta_{SS}$ anomalous events. $\eta_{Ri}$ occurring during low discharge events (here low is defined as less than $10\ \mathrm{m^3 s^{-1}}$, the approximate summer average discharge) is added back into the La Push $\eta_{SS}$, as it is likely not driven by river forcing. After $\eta_{Ri}$ was removed from the $\eta_{SS}$ signal, it is saved as a time series of river-forced water level events.

## 4.3 SR14 modifications for estuarine environments

SR14 was originally developed to simulate TWLs in a Monte Carlo sense in open-coast environments and does not have a mechanism in place for simulating the new variables of interest, river discharge (Q) and $\eta_{Ri}$. SR14 was therefore modified to include simulations of $\eta_{Ri}$ and Q at both the Calawah and Sol Duc rivers. To do this, relationships were formed with variables already simulated within the SR14 model.

High discharge events in the Sol Duc and Calawah (and therefore Bogachiel) rivers tend to occur within hours of peak wave and water level events. Due to the interrelated nature of these forcings, daily maximum estimates of Q at the Calawah River are compared to all variables simulated in the SR14 model (e.g., Hs, $\eta_{SS}$, $\eta_{NTR}$, $\eta_{MMSLA}$, etc.) to capture any dependency inherent in these processes. The most correlated variable to Q is Hs.

Similar to methods in SR14, extreme Hs and Q events at the Calawah River are determined using the Peak Over Threshold
approach, where all independent daily maximum events over a defined threshold are selected. Threshold excesses are fit to non-stationary Generalized Pareto distributions, which include seasonality as a covariate. Both variables are transformed to approximately Fréchet margins. A bivariate logistics model is then used to model the dependency between the variables. To simulate, random numbers are sampled from a uniform distribution and mapped to each variable's prescribed Fréchet





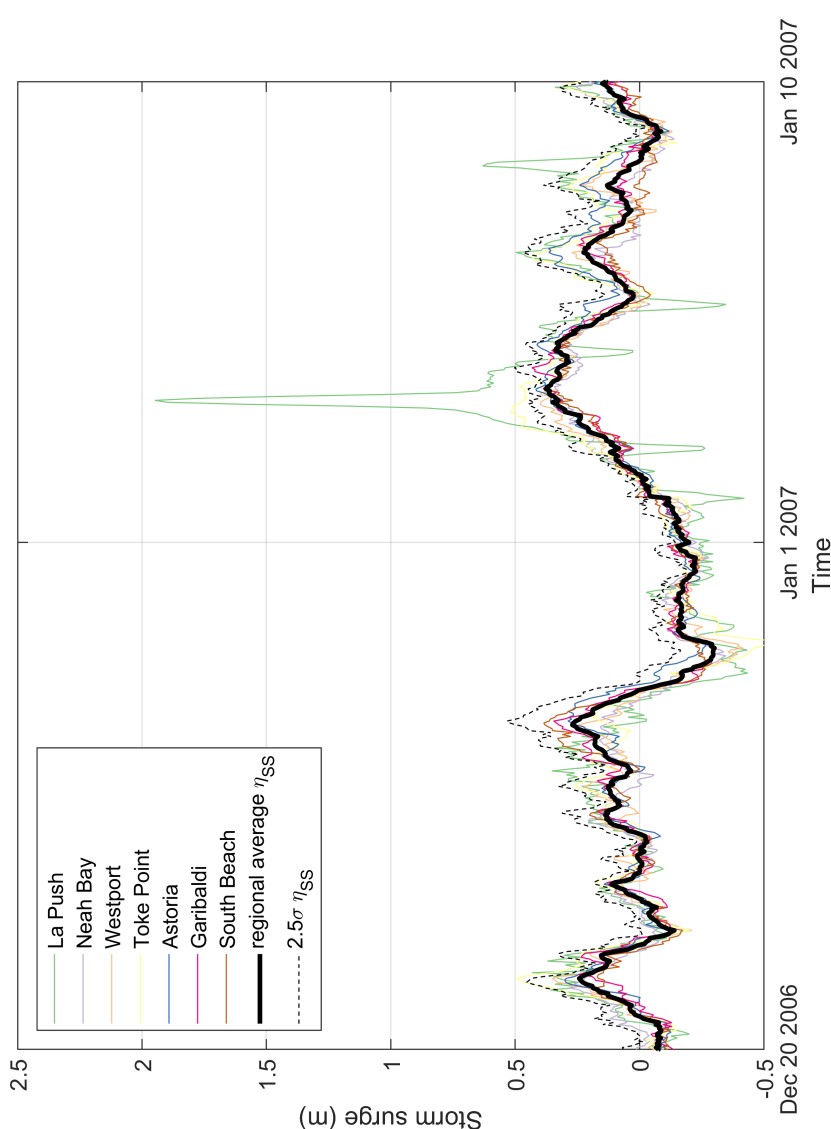

**Figure 6.** A comparison of storm surge ($\eta_{SS}$) decomposed from all tide gauges along the northern Washington to central Oregon coastline. The solid, black line depicts the regional average of all of the $\eta_{SS}$ signals, while the dashed black line represents the regional average $\eta_{SS} + 2.5*\sigma$ of all $\eta_{SS}$ in the region. When the La Push $\eta_{SS}$ exceeds the regional average $\eta_{SS} + 2.5*\sigma$ it is removed from the record and considered river influence.



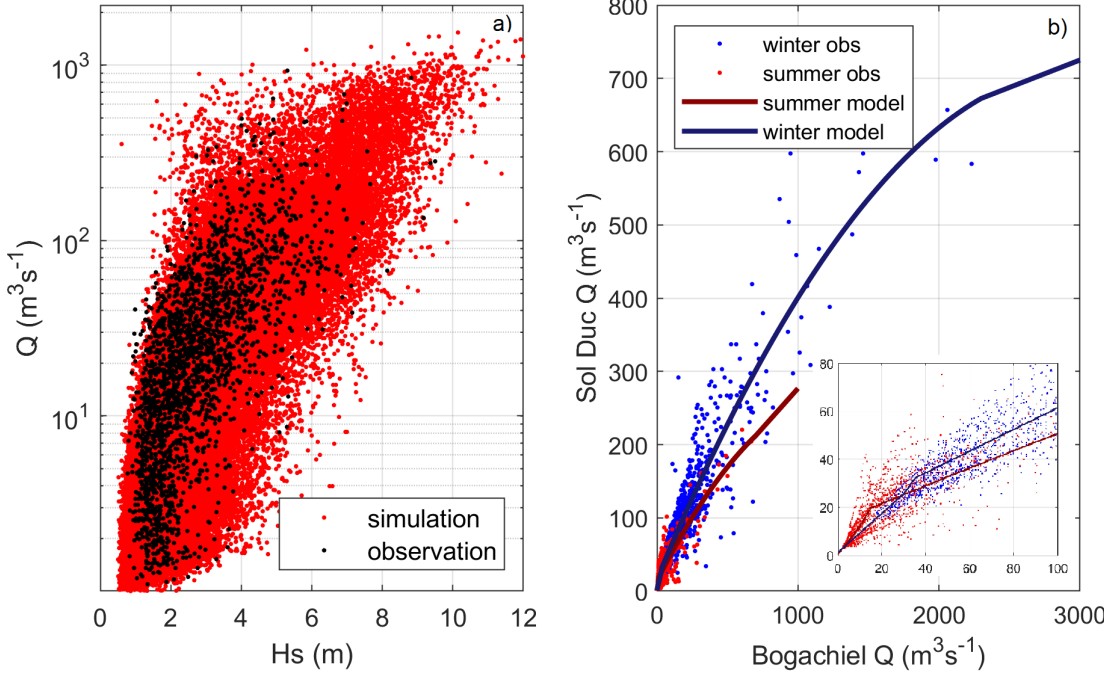

**Figure 7.** a) Joint relationship between wave height (Hs) and storm surge and discharge (Q) for the observational record (black) and one example 500 year simulation (red). b) Seasonal model fit for the probabilistic simulation of the Sol Duc River Q in relation to the Bogachiel River Q. The inset displays the model fits for discharge less than 30 $m^3s^{-1}$.

cumulative probability distribution function. Based on the probability of occurrence of the transformed value, the estimate is transformed back to the physical scale using the Generalized Pareto distribution if extreme, dependent on the variable's threshold. If not extreme, the estimate is transformed back to the physical scale using monthly-varying Gaussian copulas. This technique generates a synthetic record of Q at the Calawah River gauge that is seasonally varying, related to larger-scale

5 climate variability through wave height (essentially as a proxy for storms), and carries the same dependency between variables as the observational record (Figure 7). Q is then multiplied by 2.09 to represent inflow from both the Bogachiel and Calawah rivers. The bivariate logistic model preserves the dependency and frequency of occurrence of joint Hs-Q events in extreme and non-extreme space. This modeling technique is also used to simulate $\eta_{SS}$ in SR14.

Because discharge measurements at the Sol Duc River are highly correlated with the Calawah River ($\rho = 0.9$, $\tau = 0.83$), the

10 Sol Duc River is modeled based on a relationship with the Calawah River. Once the Calawah River is scaled to represent the Bogachiel River, estimates of Q at the Sol Duc River are related to the Bogachiel River during the summer and winter seasons. First, daily maximum Q is split into summer (May, June, July, August, September, and October) and winter (January, February, March, April, November, December) seasons. Next, two models are fit to the joint relationship between the Sol Duc River Q



(hereinafter $Q_{SD}$) and the Bogachiel River Q (hereinafter $Q_B$) each season, such that for the summer season, a best-fit linear model represents $Q_{SD}$ when $Q_B$ falls between 0-10 m$^3$s$^{-1}$, and a best-fit quadratic represents $Q_{SD}$ when $Q_B$ falls between 10 - 700 m$^3$s$^{-1}$ (Figure 7). For the winter model, a linear model is fit to $Q_{SD}$ when $Q_B$ fell between 0-30 m$^3$s$^{-1}$, and a quadratic when $Q_B$ falls between 30 - 2300 m$^3$s$^{-1}$ (Figure 7). Equally spaced bins are determined and residuals of $Q_{SD}$ from the model

fits are generated. Normal distributions are fit to the $Q_{SD}$ residuals in each bin, except for low bins (less than 30 m$^3$s$^{-1}$) where residuals are fit to exponential distributions. $Q_{SD}$ is then directly related to simulated estimates of $Q_B$; $Q_{SD}$ is first determined by fitting the prescribed model to each estimate of $Q_B$, and then a random sample is taken from the residuals per that bin and added to the model. This technique captures the joint-peaks of the river systems visible in the observed dataset, while allowing for variability in the simulated estimates.

The largest $\eta_{Ri}$ usually occur coincident with low tide. This is likely due to the competing ocean and river processes during high Q events. During high tide, riverine floodwaters are blocked from outletting to the ocean and back up in the river. As the water recedes during low tide, the river is no longer suppressed and exits through the inlet (Kumbier et al., 2018; Chen and Liu, 2014). The drainage of the river into the ocean generates high water levels at the mouth, elevating the SWL during low tide, driving a peak in the $\eta_{NTR}$. ADCIRC simulations confirm this phenomenon, as the river discharge peak is modeled exactly at

low tide (Figure 5). We are, however, most interested in the maximum daily SWL that drives flooding, which generally occurs during, or close to, the daily high tide. Modeling large peaks in $\eta_{Ri}$ that occur during low tide would therefore erroneously increase simulated estimates of the SWL occurring during high tide. Thus, instances of $\eta_{Ri}$ occurring approximately during high tide are retained and all other $\eta_{Ri}$ peaks are discarded. The resulting 155 peaks in $\eta_{Ri}$ are correlated with $Q_B$ (Figure 8).

In order to statistically simulate $\eta_{Ri}$, two linear regression models are fit to $Q_B$ and $\eta_{Ri}$, where $Q_B$ is the independent

variable. Two models rather than one are chosen because the elevation of $\eta_{Ri}$ increases and becomes more varied as $Q_B$ increases. The first linear model is fit to $Q_B$ below 190 m$^3$s$^{-1}$, and the second is fit to $Q_B$ above 190 m$^3$s$^{-1}$. Next, coarse bins ranging from 100 to 400 m$^3$s$^{-1}$ are created and the $\sigma$ of $\eta_{Ri}$ values within each bin is saved. For bins that contained less than 10 observations, observations from the previous bins were included until there were more than 10 observations per bin for $\sigma$ calculations. Finally, a 2-point running average was used to smooth the $\sigma$ from each bin to ensure continuous transitions

and avoid the edge-effects from binning a sparse dataset. After $Q_B$ were simulated using SR14, the developed modification simulates $\eta_{Ri}$ for every day in time by selecting the synthetic daily estimate of $Q_B$ and randomly sampling from a normal distribution for each $Q_B$ bin, where the distribution parameters are modeled as $\mu$ = the regression model and $\sigma$ = the standard deviation from each bin (Figure 8).

There are times of high $Q_B$ without a distinguishable $\eta_{Ri}$ in the tide gauge record, thus a model is also developed to simulate

the frequency of occurrence of $\eta_{Ri}$ as not to artificially elevate SWLs. The frequency of occurrence of $\eta_{Ri}$ is therefore defined as the percentage of time $\eta_{Ri}$ occurs in the observational record. In the observational record, $\eta_{Ri}$ occurs less than 10% of the time when $Q_B$ is less than 210 m$^3$/s, and 15 - 25% of the time when Q is between 840 and 2090 m$^3$s$^{-1}$ (Figure 8). For Q greater than 2090 m$^3$s$^{-1}$, $\eta_{Ri}$ occurs during daily maximum water levels approximately 50% of the time. Estimates of the percentage of time $\eta_{Ri}$ occurs are modeled by a best-fit cubic function to the percentage of time the values have occurred in

the record. Because there is no record of events greater than 2500 m$^3$s$^{-1}$, we represent the percentage of occurrence as 100%,





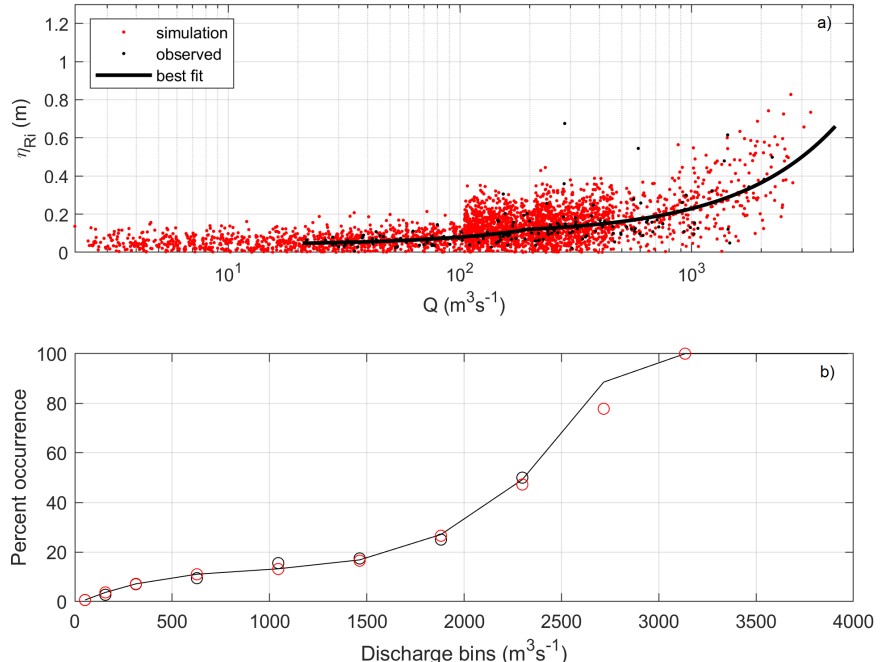

**Figure 8.** a) The relationship between the river-influenced water level ($\eta_{Ri}$) and river discharge (Q). The solid black line represents the linear fit to the observational records (black dots). b) The percentage of time $\eta_{Ri}$ occurs in the record during a specific Q. In both panels, black represents the observational record and red represents one example 500 year simulation.

because at some point, large Q events would drive $\eta_{Ri}$ to occur 100% of the time (Figure 8). The example simulation shows SR14 captures both the spread of $\eta_{Ri}$ related to Q events as well as the percentage of time of occurrence (Figure 8).

# 5 Results

The following section first provides a validation of the surrogate models by comparing along-river water levels from a specific

5 set of conditions directly modeled in HEC-RAS to along-river water levels interpolated from the surrogate models for the same set of conditions. Next, the spatial and temporal variability of the magnitude of along-river water levels and their driving conditions are examined. Finally, low probability water levels, like the 100-yr event, are extracted and their dominant drivers are evaluated and compared to the low probability water level from the 100-yr discharge or 100-yr SWL event at each transect.

## 5.1 Surrogate models

10 Approximately 3,000 Q-SWL validation scenarios are directly modeled through HEC-RAS to determine if the number of conditions used for surrogate model generation represent a large enough sample space of forcing conditions for correctly





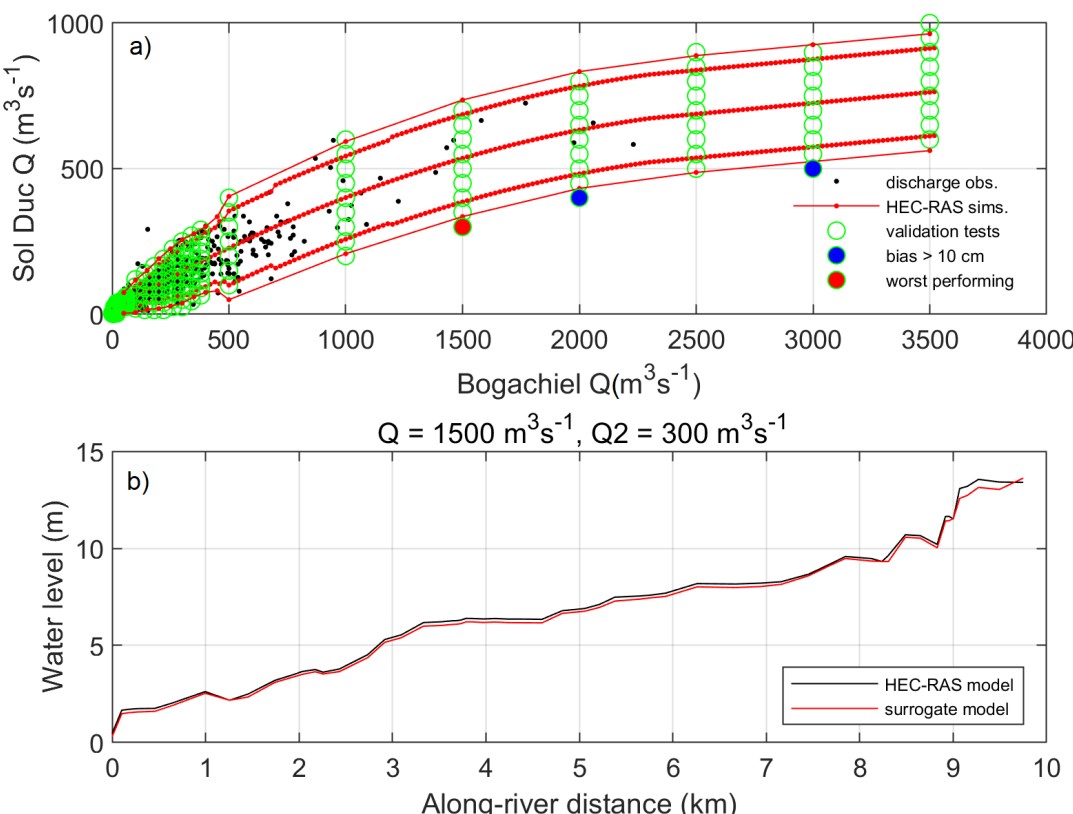

**Figure 9.** a) Modeled HEC-RAS Q boundary conditions used to generate the surrogate models (red-dotted lines) compared to the simulated conditions used for surrogate model validation (green dots). The black dots represent the observational daily max conditions, while the colored circles represent the worst-performing of the validation tests. The red and blue colored circles represent the scenarios where the interpolated water surface had a bias of over 10 cm lower than the model. b) Example along-river water level for the worst performing condition in the validation tests.





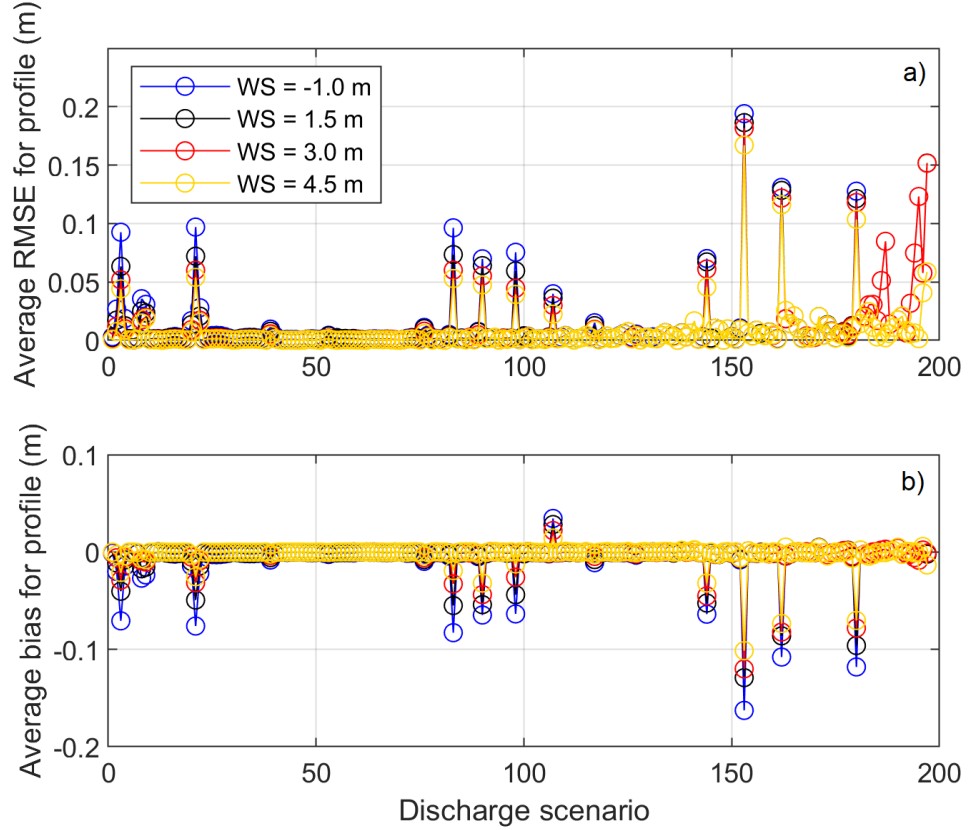

**Figure 10.** a) Average root mean square error (RMSE) and b) bias for all validation scenarios (e.g., 197 Q and 15 SWL) across four SWL scenarios. The worst-performing model (pictured in the previous figure) is discharge scenario 153.

interpolating along-river water levels. The validation scenarios are chosen to cross through both HEC-RAS modeled and unmodeled conditions (Figure 9). Across all validation scenarios, the average root mean square error (RMSE) between the directly-modeled and surrogate model-generated water level is 1 cm. Only about 1.5% of the validation scenarios have a bias greater than 10 cm, and the largest RMSE at any transect is 20 cm across all water level scenarios (Figure 10). The worst

5   represented scenarios occur during high Bogachiel River Q events paired with low Sol Duc River Q and low SWL events. However, even during these cases, the differences between the surrogate model-interpolated and directly modeled water levels are small (Figure 9). The main research interest here is extreme water levels, and the conditions driving low probability return level events rarely fell around the scenarios with the highest bias.

## 5.2 Temporal variability in along-river water levels

10   Similar to the driving boundary conditions of SWL and Q, seasonal variability exists in the elevation of along-river water levels. The highest elevation water level occurs during the winter (here defined as December, January, and February), while





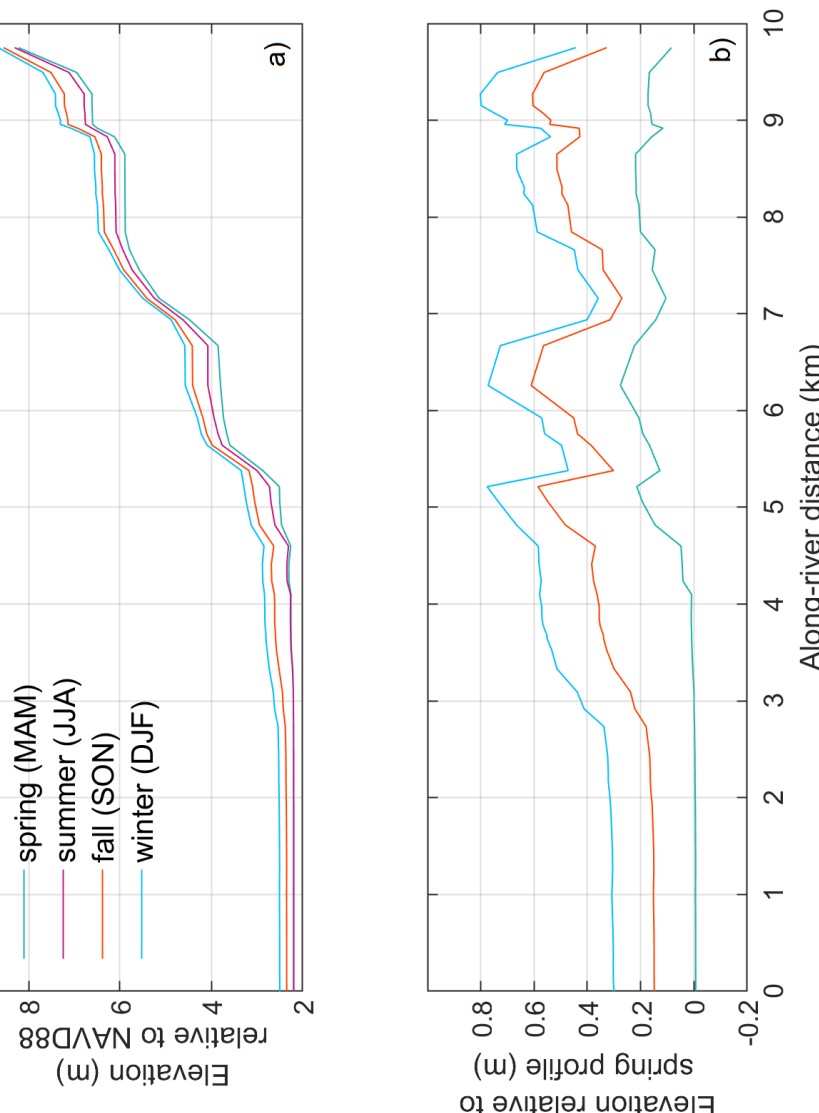

**Figure 11.** a) Variability of along-river water levels averaged over summer (JJA), fall (SON), winter (DJF) and spring (MAM). b) The difference between the fall, winter, and summer and the spring along-river water level.

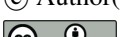



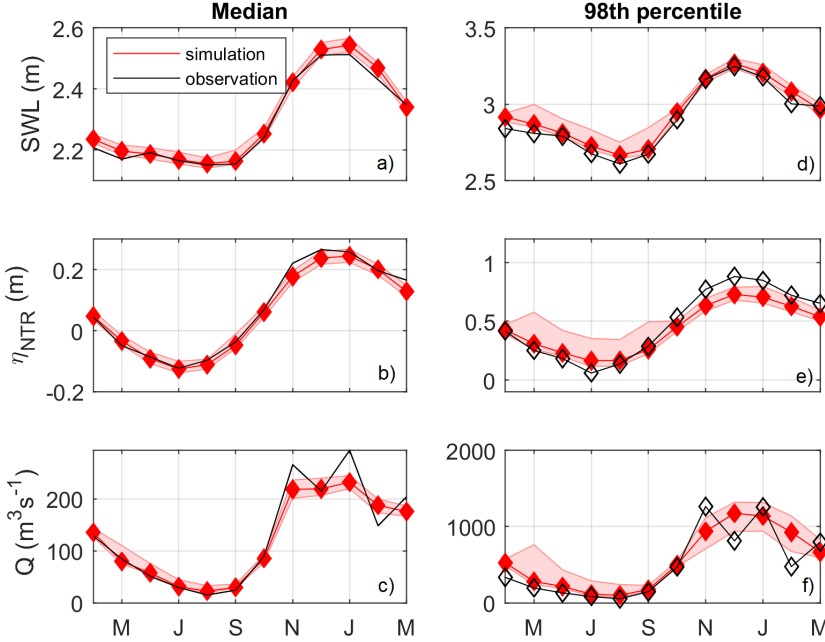

**Figure 12.** Left) Observational (black) and simulated (red) monthly median still water level (SWL), non-tidal residual ($\eta_{NTR}$), and discharge (Q). Right) Observational (black) and simulated (red) monthly 98th percentile of the SWL, Q, and $\eta_{NTR}$. Red shading indicates the bounds value from each simulation.

the lowest elevation water level occurs during the spring (March, April, May) ( Figure 11). The spring profile is on average (maximum difference) 50 cm (84 cm) lower than the winter profile, 33 cm (63 cm) lower than the fall (September, October, and November) profile, and 3 cm (12 cm) lower than the summer (June, July, and August) profile (Figure 11). The difference between seasonal profiles is nonlinear upstream, and certain sections of the river have larger changes in elevation between

5    months (Figure 11). However, this variation becomes relatively linear downstream of river km 3.

The seasonal variability of the along-river water level is driven by the seasonality of the forcings, which are well represented in the simulations compared to the observations (Figure 12). The median Q of the Quillayute (combined Sol Duc and Bogachiel Q) is approximately 200 m$^3$s$^{-1}$ higher in winter months than summer months (Figure 11). This cyclical variability is also depicted in the monthly median SWL and $\eta_{NTR}$. Winter $\eta_{NTR}$ is approximately 40 cm higher than summer $\eta_{NTR}$, which is

10    also reflected in the SWLs (a and b, Figure 12). The 98th percentile of Q, SWL, and $\eta_{NTR}$ have a similar seasonal variability as median conditions (Figure 12).





## 5.3 Probabilistic spatially-varying extreme water levels

Using the count-back method, water level return level events at each transect are extracted for all 70 500 year long simulations representing present-day climate from 1980-2016. This methodology thus provides both an estimate of the average water level return level as well as the uncertainty around that value. The magnitude of along-river water level return level events are

between 2 and 10 m, with peaks near 1, 3 and 9 km (Figure 13). While the peaks in water level return level events occur at similar locations, the difference between water level return level events spatially varies moving upriver. For example, at river km 1, the difference between the average annual and 100-yr event is approximately 50 cm, whereas at river km 9.5, the difference between the two events is 2 m (Figure 13).

    The many realizations of joint SWL-Q allows for the investigation of the fluvial and oceanographic processes driving the

magnitude of water level return level events. Panels c and d in Figure 13 displays the average condition forcing the water level return level for the annual, 25, 100, and 500-yr event. Between river km 0 and 1.5, the average SWL driving the water level return level event is constant and then gradually decreases over a 1 km zone by approximately 50 cm. On the other hand, the average Q driving the water level return level event gradually increases by approximately 2000 $m^3s^{-1}$ over river km 0 - 3 and then is fairly constant from river km 3 to 10 (Figure 13). Compared to the univariate return level forcings, we find that the

stretches of river that display constant SWL or Q forcing approximate the univariate return level event such that the 100-yr SWL does indeed cause the 100-yr water level in the lower river near the ocean outlet, while the 100-yr Q event drives the 100-yr water level along river km 3 - 10 (grey dashed lines, Figure 13). However, between river km 1.5 - 2.5 a flood transition zone is present, where neither the SWL return level or the Q return level drives the water level return level. This is consistent across all return level events, regardless of likelihood. This is further evidenced by investigating the SWL and Q conditions that

drive the annual and 100-yr event at specific along river transects (Figure 14). At the river mouth, the annual water level event occurs during Q ranging from 20 - 3200 $m^3s^{-1}$ and SWLs that vary by only 10 cm. Moving upstream to river km 2, which lies in the flood transition zone, the annual event is driven by both high SWL occurring during low Q and low SWL occurring during high Q. By river km 4, the annual event is forced by the univariate, annual Q event (Figure 14). This pattern is similar for the 100-yr event at all transects but with higher magnitude SWL and Q conditions.

## 6 Discussion

The hybrid model developed in this study, which combines statistical simulations with a physics-based model, provides a novel approach for probabilistically evaluating the conditions that drive extreme water levels, not only at a tide gauge, but also miles upriver. The ability to simulate hundreds of thousands of combinations of Q and SWL events allows for a robust estimate of resulting along-river water levels, which numerical models alone are unable to consider due to large computational

expenses. While some of our modeling techniques are specific to this location, the overall framework for combining statistical and physics-based models is general enough for use in coastal locations throughout the globe where flooding arises from compounding processes.




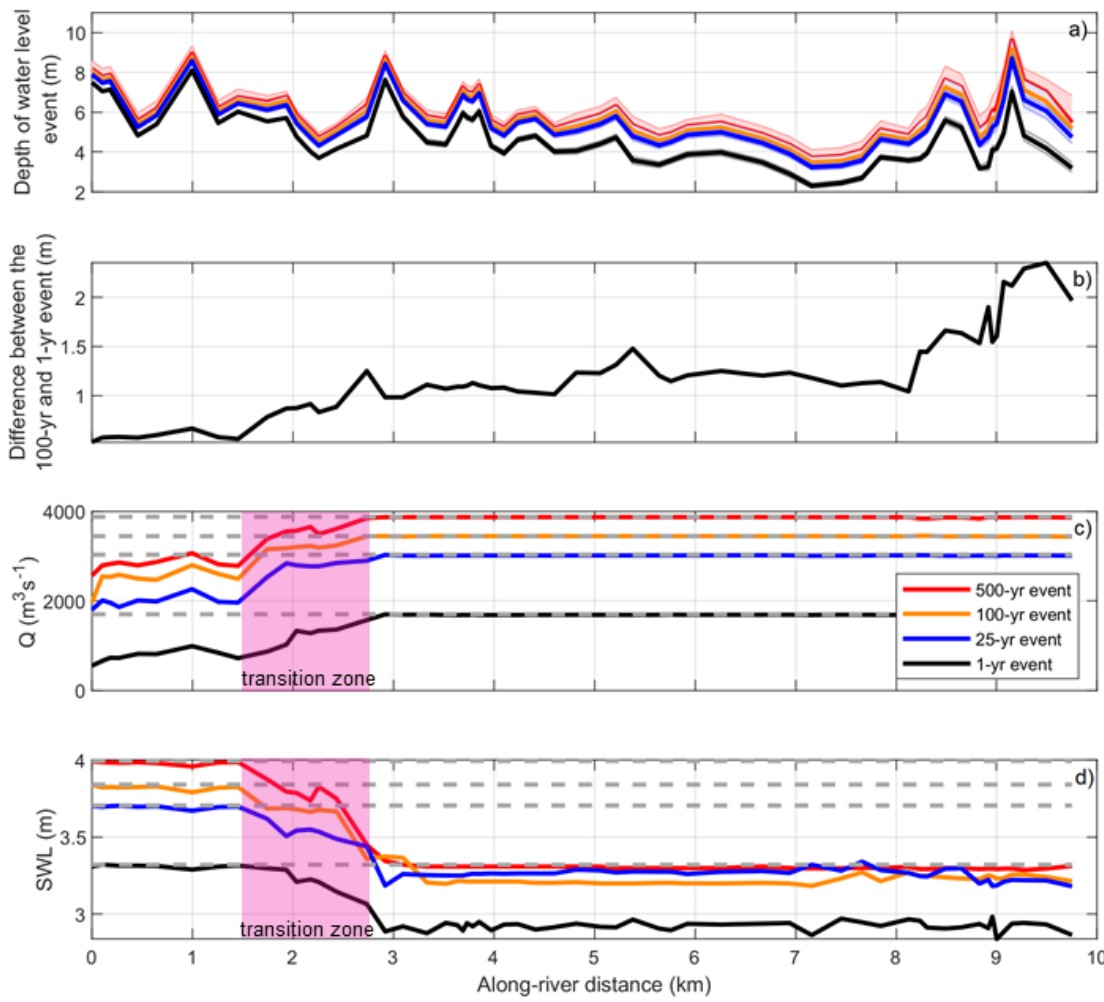

**Figure 13.** a) The average along-river water level return level at each transect for all 70 probabilistic simulations. b) The along-river difference between the average annual and 100-yr event. The average forcing condition driving the response-based return level at each river transect where c) displays the Quillayute Q scenario driving low probability water levels and d) displays the SWL scenario driving low probability water levels. The grey dashed lines depict the event-based return level, where the low probability water level would be modeled based off , for example, the low-probability discharge. Red, orange, blue, and black lines represent the 500, 100, 25, and annual return level event. In panels c and d, the pink shaded area represents a transition zone, where neither event drives the water level.



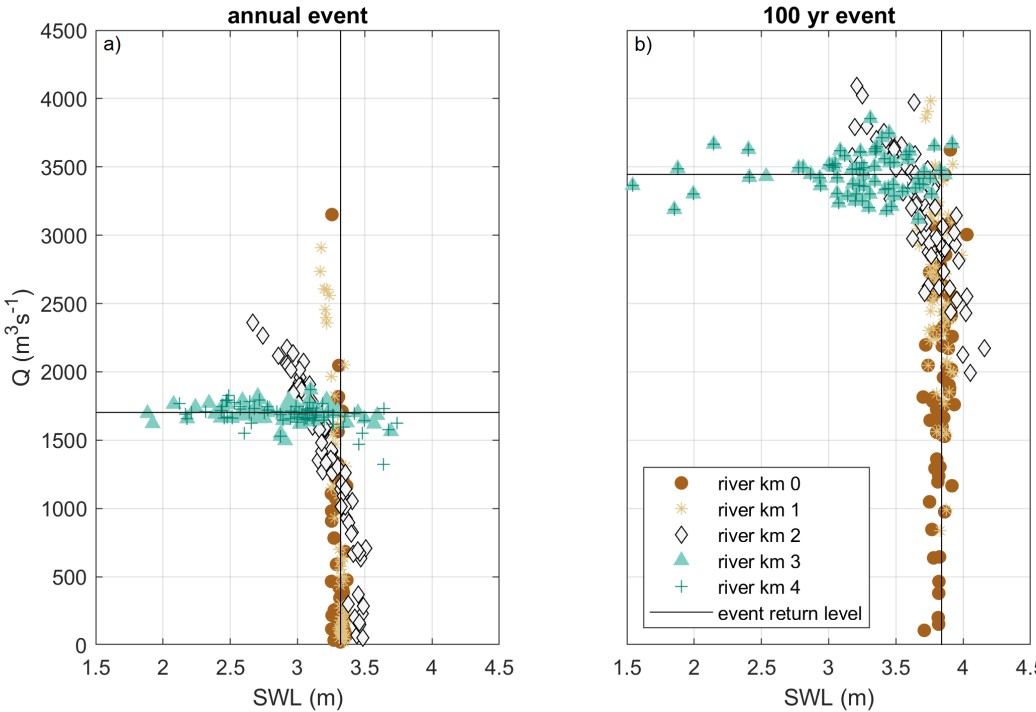

**Figure 14.** The individual Q or SWL event driving the a) annual and b) 100-yr event at specific along-river locations.

The decomposition of the SWL into low and high frequency signals, including a river-influenced component, helps characterize the importance of physical processes in various regional settings. This is especially important in locations like the US West coast, where the steep, narrow continental shelf prevents wind and pressure driven storm surge from being overwhelmingly large (Allan et al., 2011). The influence of the river signal in the tide gauge is directly related to the setting of our study

site. The estuary is relatively small and narrow with the river discharging directly into the ocean. This is dissimilar to other tide gauges in the region which are located in larger estuaries, situated away from river input. Estuaries typically exhibit wave, tide, or river-dominant morphology, based on the relative energy of each process (Dalrymple et al., 1992). The Quillayute River outlets directly to a high wave energy environment and has a small estuary volume compared to its river input volume. The steep catchment of the mountainous environment means a short response time for rainfall, therefore producing peak discharges

temporally similar to peak storm-induced still water levels, allowing for interaction between the two. In contrast, water level elevations with large estuary volume compared to river discharge are less influenced by fluvial processes. Furthermore, a larger estuary may experience variability in the water surface elevation due to wave-induced setup and/or other local storm-induced processes (Cheng et al., 2014; Olabarrieta et al., 2011), which may further dampen the influence of a river signal.

Defining compounding extreme events based on a more complete probability space of jointly-occurring conditions has

been described in open coast settings (Serafin et al., 2017), however this is the first application to riverine environments.



This research confirms the presence of an oceanographic-fluvial transition zone, where traditional, univariate methodologies for defining return level events are insufficient for defining water level return levels. Between river km 1.5 and 2.5, we find that a range of SWL and Q events drive all return level events, and neither the univariate SWL or Q return level drives the water level. A similar flood zone transition was recently modeled numerically, and albeit for a single event, physically demonstrated the importance of including multiple variables to reproduce accurate flooding (Bilskie and Hagen, 2018). Thus, flood hazard assessments on systems with multivariate forcings may misrepresent water level elevations for low probability events if only univariate variables are modeled. This has large implications for characterizing the risk to flooding, especially in the context of mapping flooding hazards. Furthermore, we show that return level water levels can occur over a range of combined extreme and non-extreme forcing in the flood transition zone. This illustrates that in order to properly understand the impacts of compounding flooding, more than just design scenarios need to be considered for the proper assessment of risk.

Many of our results can be explained by dynamics that occur during interacting ocean and river flows. For example, a coincidence of high SWL and peak river discharge may induce blocking, where river-induced water levels are trapped upstream and either flood overbank or outlet to the ocean when water recedes (Kumbier et al., 2018; Chen and Liu, 2014). While our ADCIRC simulation confirms the presence of this effect by matching the peak storm surge at low tide, our hybrid methodology only models steady flow scenarios. Thus, with co-occurring daily maximum SWL and discharge, we may miss certain dynamics important for flooding over unsteady conditions. At low tide, a high river discharge may promote drainage of the floodwater into the ocean (Kumbier et al., 2018), increasing water levels for days at a time and prolonging exposure to flooding. Furthermore, interactions between storm surge and river discharge may increase the overall elevation of the residual (Maskell et al., 2013).

Because sea level rise, along with other changes to the climate, will exacerbate the compounding effects of flood drivers (Moftakhari et al., 2017; Wahl et al., 2015), it is also important to consider the impact of changes to processes driving flooding events in the future (Zscheischler et al., 2018). By 2100, the likely range of relative sea level rise in the La Push area is projected to be between 18 and 80 cm, considering vertical land motion and high and low emissions scenarios (Miller et al., 2018). The western Olympic Peninsula is projected to experience increased winter precipitation (Mote et al., 2013; Halofsky et al., 2011) which could subsequently increase either the frequency or intensity of high Q events along the Quillayute River. While we have characterized the spatial variability in extreme water levels in the present-day, there is a high likelihood changes in the future climate will shift the importance of these interacting processes.

## 7 Conclusions

This research illustrates the importance of considering a large number of forcing conditions to model compounding processes when evaluating extreme water levels. Here we find that in coastal settings, river discharge can be an important driver of high water levels measured in a tide gauge. We also find that the univariate, forcing-driven return level event, like the 100-yr discharge, does not always match the response return level, like the 100-yr water level. Furthermore, when processes compound, the low probability water level may be driven by events that are not extreme themselves. Probabilistic techniques allowing for





the analysis of thousands of combinations of events not captured in the observational record provides a robust characterization of where river, ocean, or the combination of the two, may be important for generating extreme events.

Overall, the hybrid merging of a statistical and numerical model provides a methodology for better understanding the drivers of flooding along the length of a river. While our model does not actively resolve the physical interaction of river and oceano-
graphic flow, it develops an approach for characterizing and extracting river-influenced water levels measured at tide gauges while robustly modeling the drivers of extreme along-river water levels. Understanding the drivers of flooding events now and into the future will ultimately increase the preparedness of the community of La Push.

## 8    Data availability

Data can be made available by the authors upon request.

## 9    Author contribution

The study and methodology were conceived by KAS and PR. KAS carried out the analyses, produced the results, and wrote the manuscript under the supervision of PR. KAP carried out the analyses and produced the results of the ADCIRC simulations. KAP also developed the topography/bathymetry DEM as well as the geometric files for use in HEC-RAS. KAS, PR, KAP, and DFH all contributed by generating ideas, discussing results, and manuscript editing.

## 10    Competing interests

The authors declare that they have no conflict of interest.

*Acknowledgements.* Tide gauge records are available through the National Oceanic and Atmospheric Administration (NOAA) National Ocean Service (NOS) website and river discharge is available through the U.S Geological Survey (USGS) National Water Information System (https://waterdata.usgs.gov/wa/nwis/rt). Bathymetric and topographic data were obtained from NOAA's Elevation Data viewer (DEMs). This
work was funded by the NOAA Regional Integrated Sciences and Assessments Program (NA15OAR4310145) and a contracted grant with the Quileute tribe.



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
