# Peer review of "What's streamflow got to do with it? A probabilistic simulation of the competing oceanographic and fluvial processes driving extreme along-river water levels"

_Natural Hazards and Earth System Sciences, 2018_

## Referee Comment (RC1) · Anonymous Referee #1 · 4 Mar 2019

General comments Serafin et al present a new framework for examining the joint influence of several coastal and riverine processes on water levels in estuarine environments, and show very clearly that the 100-yr ocean or 100-yr streamflow event does not always produce the 100-yr along-river water level. It is a novel piece of work using a clever methodological framework, resulting in an analysis that can assesses non-stationary water levels from a multivariate joint distribution and truly decompose coastal water levels. As such, I believe that the research forms an important contribution to the increasingly important field of compound flood risk assessment. The

manuscript is well written in terms of language, but parts of it feel to long or could be helped by restructuring. There are also some specific methodological issues that require further explanation, as described in the following review. Nevertheless, if these can be sufficiently responded to, I believe that this paper would provide a very valuable addition to the literature.

Main comments

The introduction is generally well written and reviews most of the relevant literature. However, some important concepts for the paper are not fully introduced or defined. For example, a formal definition of compound flooding is missing. On page 2, line 20 (and also later at page 25, line 15) the authors imply that probabilistic simulations of water levels have not yet been done considering ocean and onland processes, and that this has only been done for specific events. However, Bevacqua et al (2017) van den Hurk et al. (2015), and Couasnon et al. (2018) have used probabilistic simulations. The current paper certainly adds value to the research carried out in those studies, but it would be prudent to mention them and how the current study advances.

In terms of the overall structure, the methods section (section 3, but also parts of section 4) are sometimes difficult to follow. The really interesting part here is the new overall framework. However, this overall framework sometimes gets lost in the details of the various specific models used, which can be rather lengthy (e.g. the part on HEC-RAS). It would be beneficial to the reader to highlight the overall methodological framework more clearly at the start of the methods section, for example with a flowchart. This would highlight more clearly the major novelty of this paper. It is of course also necessary to give details of the various components used for each part of the framework, such as HEC-RAS. But by emphasizing more the framework, it would be clear that one could also use the overall assessment framework with other hydrodynamic models, if one wished to do so.

Following on, it may help to move some of the details to Supplementary Information.

General background information about setting up HEC-RAS can be shortened, and the essential parts for this study could be moved to supplementary information. This would improve overall readability of this section.

Related to the previous comments on structure, the part on HEC-RAS model validation (3.2.1) seems out of place in the methods section. It could be moved to the section on validation or in my opinion better to still to supplementary information.

My main methodological concern relates to the use of steady flow simulations. As the authors state themselves in the discussion, the steep catchment of the mountainous environment means a short response time for rainfall. It also calls into question the validity of using steady-state flow for the analysis. I would like the authors to explain this choice and explain what it means for the overall results? Has there been any sensitivity assessment of the results compared to an unsteady state simulation, for example?

It is not clear what Manning's coefficients are used on the flooplains. It is stated that they are estimated using 2011 Land Cover data from the Western Washington Land Cover Change Analysis project (NOAA, 2012) and visual inspection of aerial imagery. But what values were selected for different land use classes? Moreover, on page 8, line 20 the Manning coefficient of "0.005" is very low and not really representative of natural river states. Is there a specific reason for this?

How are the high water level events constructed? The possible presence of autocorrelation in the data is not mentioned – it would be good to test for this or report the results of such a test if it has been done already.

Other suggestions

Figure 13: the grey dashed lines presumably belong to the 4 different return periods shown – it would be easier for the reader to use the same colours (but dashed) instead of grey.

Caption of figure 13: "the pink shaded area represents a transition zone, where neither event drives the water level". The last part is not clearly phrased. Do you mean the zone where the water level is not driven by either the coastal or river drivers alone?

Page 26, lines 14-15: "At low tide, a high river discharge may promote drainage of the floodwater into the ocean (Kumbier et al., 2018), increasing water levels for days at a time and prolonging exposure to flooding". Why would a low tide that promotes drainage to the ocean lead to increased water levels? Would the opposite not lead to backwater effects?

In the abstract it is stated that "Understanding the relative forcing of extreme water levels along an ocean-to-river gradient will better prepare communities within inlets and estuaries for the compounding impacts of various environmental forcing". A similar statement can be found in the conclusions. I feel that this requires more nuance. There are many steps that would be needed to make these (important) scientific insights usable by a local community for preparing themselves.

Page 17-line 14-15: "ADCIRC simulations confirm this phenomenon, as the river discharge peak is modeled exactly at low tide (Figure 5)". I find it hard to see that when looking at Figure 5. Maybe help the reader a bit more? For me it seems more to be at high tide but maybe there is something I am missing.

Textual changes

Page 3, line 30. Change "…experiencing relative sea level rates of…" to "…experiencing relative sea level change rates of…" (similar comment in line 31).

Page 8, lines 10-11: add "in most cases".

Page 8, line 30 (and the rest of the text): where is Toke Point tide gauge on Figure 1?

Page 11, line 12. Change "periosd" to "period"

Page 14, line 13. Change "subsituting" to "substituting"

Page 23, line 19: suggest to remove "regardless of the likelihood" (it is already in the return level events?)

Page 23-line 5 and 8: add "a" and "b" to Figure 13 to help the reader.

References not mentioned in manuscript

Bevacqua E, Maraun D, Haff I H,WidmannM and Vrac M 2017 Multivariate statistical modelling of compound events via pair-copula constructions: analysis of floods in Ravenna (Italy) Hydrol. Earth Syst. Sci. 21 2701-2723.

Couasnon A, Sebastian A and Morales-Nápoles O 2018 A Copula-based bayesian network for modeling compound flood hazard from riverine and coastal interactions at the catchment scale: An application to the houston ship channel, Texas. Water, 10, 9, 1190   Van den Hurk B, van Meijgaard E, de Valk P, van Heeringen J and Gooijer J 2015 Analysis of a compounding surge and precipitaiton event in the Netherlands Environ. Res. Lett. 10, 035001

---

## Referee Comment (RC2) · Anonymous Referee #2 · 5 Mar 2019

The paper overall presents a good contribution, however, it needs some work Some concepts are not clear, and the reader is left 'guessing' about their meaning. for example, in the introduction, a reader is not aware of what 'bivariate or multivariate processes' are, thus they can't understand the challenge in trying to identify them or study them. My major concerns are related to the method section, that currently needs much improvement. In its present state, it is much too long in some parts, and not enough clear on the overall framework, which is the added value of this work. There is far too much description of known elements, such as HEC-Ras, for example, and not enough

clarity on the proposed approach. Also, it is not too clear if chapter 4 is a method or a discussion of results. As a consequence, it is very hard to understand the discussion of the results.

---

## Author Comment (AC1) · 14 May 2019

"The paper overall presents a good contribution, however, it needs some work Some concepts are not clear, and the reader is left 'guessing' about their meaning. for example, in the introduction, a reader is not aware of what 'bivariate or multivariate processes' are, thus they can't understand the challenge in trying to identify them or study them. "

[Figure]

Response: We thank the reviewer for pointing out the lack of contextual information in the initial submission. Bivariate and multivariate processes are processes that occur from two or multiple variables, respectively. In coastal environments, multiple processes like waves, tides, storm surge, and river discharge, may combine to drive an extreme flood event. We have improved the clarity of our descriptions of multivariate and bivariate processes by removing the sentence driving confusion (Page 1, Line 21-22 original manuscript) while introducing a formal definition of a compound event in the first line of the introduction, Page 1, Lines 16-21, "Coincident or compound events are a combination of physical processes in which the individual variables may or may not be extreme, however the result is an extreme event with a significant impact (Zscheischler et al., 2018, Bevacqua et al., 2017, Wahl et al., 2015, Leonard et al., 2014). Flooding is often caused by compound events, where multiple factors impact both open coast and estuarine environments. Storm events, for example, often generate concurrently large waves, heavy precipitation driving increased streamflow, and high storm surges, making the relative contribution of the actual drivers of extreme water levels difficult to interpret." We have also added a brief description to the abstract, Page 1, Lines 1 -2, "Extreme water levels generating flooding in estuarine and coastal environments are often driven by compound events, where many individual processes such as waves, storm surge, streamflow, and tides coincide." We hope that this revision will help readers to understand the types of events we are focused on understanding.

"My major concerns are related to the method section, that currently needs much improvement. In its present state, it is much too long in some parts, and not enough clear on the overall framework, which is the added value of this work. There is far too much description of known elements, such as HEC-Ras, for example, and not enough clarity on the proposed approach. Also, it is not too clear if chapter 4 is a method or a discussion of results. As a consequence, it is very hard to understand the discussion of the results."

Response: We agree that the amount of detail presented in the original manuscript

may have added unnecessary length and detracted from the main value of the paper and point out that Reviewer 1 had a very similar comment. Therefore, in the revised manuscript, we have moved the sections describing the HEC-RAS model domain setup, validation and calibration to the Supplemental Information. We also have moved the section describing the tide gauge merging and removal of the river-influenced water levels to the Supplemental Information. Section 4 in the original manuscript was difficult to interpret, so we merged the text from this section in with methods, results, and discussion sections in the revised manuscript in a fluid way. We have also added a schematic of the hybrid-modeling framework (Figure 3, revised manuscript and below), to help to clarify and emphasize the overall framework for readers.

References:

Bevacqua, E., Maraun, D., Hobæk Haff, I., Widmann, M. and Vrac, M., 2017. Multivariate statistical modelling of compound events via pair-copula constructions: analysis of floods in Ravenna (Italy). Hydrology and Earth System Sciences, 21(6), pp.2701-2723.

Leonard, M., Westra, S., Phatak, A., Lambert, M., van den Hurk, B., McInnes, K., Risbey, J., Schuster, S., Jakob, D. and Stafford Smith, M., 2014. A compound event framework for understanding extreme impacts. Wiley Interdisciplinary Reviews: Climate Change, 5(1), pp.113-128.

Wahl, T., Jain, S., Bender, J., Meyers, S.D. and Luther, M.E., 2015. Increasing risk of compound flooding from storm surge and rainfall for major US cities. Nature Climate Change, 5(12), p.1093.

Zscheischler, J., Westra, S., Hurk, B.J., Seneviratne, S.I., Ward, P.J., Pitman, A., AghaKouchak, A., Bresch, D.N., Leonard, M., Wahl, T. and Zhang, X., 2018. Future climate risk from compound events. Nature Climate Change, p.1.

**Hydraulic Model**
(e.g., HEC-RAS)

**Probabilistic
Simulations of
Boundary Conditions**
(e.g., SR14)

*Water Surface
Elevation*
*(at each transect
for a number of
boundary conditions)*

*pass synthetic
boundary conditions
into surrogate models*

**Surrogate Models**
(at each transect)

*Probabilistic
Estimates of Water
Surface Elevation*
*(at each transect)*

**Fig. 1.** Schematic of hybrid physical-statistical modeling technique. Models are portrayed as squares, while circles portray model outputs.

---

## Author Comment (AC2) · 14 May 2019

General comments

"Serafin et al present a new framework for examining the joint influence of several coastal and riverine processes on water levels in estuarine environments, and show very clearly that the 100-yr ocean or 100-yr streamflow event does not always produce the 100-yr along-river water level. It is a novel piece of work using a clever methodolog-

ical framework, resulting in an analysis that can assesses non-stationary water levels from a multivariate joint distribution and truly decompose coastal water levels. As such, I believe that the research forms an important contribution to the increasingly important field of compound flood risk assessment. The manuscript is well written in terms of language, but parts of it feel to long or could be helped by restructuring. There are also some specific methodological issues that require further explanation, as described in the following review. Nevertheless, if these can be sufficiently responded to, I believe that this paper would provide a very valuable addition to the literature."

Response: Thank you!

Main comments

"The introduction is generally well written and reviews most of the relevant literature. However, some important concepts for the paper are not fully introduced or defined. For example, a formal definition of compound flooding is missing. On page 2, line 20 (and also later at page 25, line 15) the authors imply that probabilistic simulations of water levels have not yet been done considering ocean and onland processes, and that this has only been done for specific events. However, Bevacqua et al (2017) van den Hurk et al. (2015), and Couasnon et al. (2018) have used probabilistic simulations. The current paper certainly adds value to the research carried out in those studies, but it would be prudent to mention them and how the current study advances."

Response: We thank the reviewer for pointing out this oversight. We have added a formal definition of a compound event in the first line of the introduction, Page 1, Lines 16-17, "Coincident or compound events are a combination of physical processes in which the individual variables may or may not be extreme, however the result is an extreme event with a significant impact (Zscheischler et al., 2018, Bevacqua et al., 2017, Wahl et al., 2015, Leonard et al., 2014)." We have also added a brief description to the abstract, Page 1, Lines 1 -2, "Extreme water levels generating flooding in estuarine and coastal environments are often driven by compound events, where many individual

processes such as waves, storm surge, streamflow, and tides coincide."

We thank the reviewer for noting that our writing seemed to imply that we were the first to produce probabilistic simulations of discharge and coastal water level events. Our intent was to highlight the novelty of the hybrid methodology merging physical and statistical models for return level analysis – which at the time the manuscript was completed and submitted was a novel application. However, after submission, Couasnon et al., 2018 and Moftakhari et al., 2019 published complementary frameworks. We thank the reviewer for suggesting the additional references, and have included the following text in our introduction to highlight the variety of previous studies, Page 2, Lines 22 – 28, "On the other hand, statistical models allow for the investigation of compound water levels through the simulation of combinations of dependent events which may not have been physically realized in observational records (Bevacqua et al., 2017, van den Herk et al., 2015). In addition, researchers have recently begun to generate hybrid models that link statistical and physical modeling approaches for understanding compound flood events (Moftakhari et al., 2019, Couasnon et al., 2018). Similar to the results solely from hydrodynamic and hydraulic models, statistical and hybrid modeling strategies show that simplifications of the dependence between multiple forcings may lead to an underestimation of flood risk."

"In terms of the overall structure, the methods section (section 3, but also parts of section 4) are sometimes difficult to follow. The really interesting part here is the new overall framework. However, this overall framework sometimes gets lost in the details of the various specific models used, which can be rather lengthy (e.g. the part on HEC-RAS). It would be beneficial to the reader to highlight the overall methodological framework more clearly at the start of the methods section, for example with a flowchart. This would highlight more clearly the major novelty of this paper. It is of course also necessary to give details of the various components used for each part of the framework, such as HEC-RAS. But by emphasizing more the framework, it would be clear that one could also use the overall assessment framework with other hydrodynamic models, if

one wished to do so.

Following on, it may help to move some of the details to Supplementary Information. General background information about setting up HEC-RAS can be shortened, and the essential parts for this study could be moved to supplementary information. This would improve overall readability of this section. Related to the previous comments on structure, the part on HEC-RAS model validation (3.2.1) seems out of place in the methods section. It could be moved to the section on validation or in my opinion better to still to supplementary information."

Response: We thank the author for the suggestions on how to better emphasize our framework. We have added a flowchart schematic that we hope better explains our hybrid modeling technique in the revised manuscript (Figure 3 in the manuscript, below as Figure 1). We agree that the amount of detail presented in the original manuscript may have added unnecessary length and detracted from the main value of the paper. In the revised manuscript, we have moved the sections describing the HEC-RAS model domain setup, validation and calibration to the Supplemental Information. We also have moved the section describing the tide gauge merging and removal of the river-influenced water levels to the Supplemental Information. We streamlined many of the sections and believe we have improved the overall readability of the manuscript.

"My main methodological concern relates to the use of steady flow simulations. As the authors state themselves in the discussion, the steep catchment of the mountainous environment means a short response time for rainfall. It also calls into question the validity of using steady-state flow for the analysis. I would like the authors to explain this choice and explain what it means for the overall results? Has there been any sensitivity assessment of the results compared to an unsteady state simulation, for example?"

Response: When originally considering merging the statistical and physical model, we started with steady flow scenarios in order to keep our simulations as simple as possible. With millions of combinations of boundary conditions, admitting discharge as a function of time would add another level of complexity to the modeling framework. However, as the reviewer mentions and as we state in the manuscript, the response time can be short, with the river rising to peak flow over 1-2 days. We completed a sensitivity assessment of steady versus unsteady simulations for a variety of hydrographs and stationary downstream boundary conditions. Figures below compare water surface elevations from the steady flow simulation of the peak discharge condition in each hydrograph to water surface elevations during the peak discharge condition from an unsteady flow simulation. Below we present results from a low, average, and extreme flow scenario (Figures 2, 4, 6, respectively).

Our results show that for many along-river locations, the steady flow approximation is reasonable. Average (standard deviation) differences between water surface elevations are 5 cm (20 cm), 7 cm (11 cm), and 40 cm (76 cm) (Figures 3, 5, 7). Large percent differences between steady and unsteady simulations are due to differences between very small numbers (e.g., Figure 7). These results show that at specific locations, unsteadiness is indeed an area of improvement, but it is currently outside the scope of what has been done.

Our methodology at this point is not intended to model any specific event perfectly, but instead to understand locations where compounding SWL and Q are statistically likely to occur together and potentially generate flooding events. This technique can also provide useful information for choosing appropriate boundary conditions for the modeling of unsteady flow scenarios. Finally, as mentioned in the manuscript, the existence of a longitudinal profile of the water surface elevation, which we were able to recreate using steady flow simulations, provided some confidence in our selection of steady flow simulations.

"It is not clear what Manning's coefficients are used on the floodplains. It is stated that they are estimated using 2011 Land Cover data from the Western Washington Land Cover Change Analysis project (NOAA, 2012) and visual inspection of aerial imagery.

But what values were selected for different land use classes? Moreover, on page 8, line 20 the Manning coefficient of "0.005" is very low and not really representative of natural river states. Is there a specific reason for this?"

Response: The Manning's coefficients used over the floodplain ranged from 0.04 (cleared land with tree stumps) - 0.1 (heavy stands of timber/medium to dense brush). These values were extracted from Table 3-1 in Brunner, 2016. For the channel, Manning's coefficients were calibrated on a transect by transect basis to determine the best-fitting longitudinal water surface profile compared to the measured data. This technique of optimizing the Manning's coefficient is widely used in the literature when observational water surface profiles exist (e.g., Wasantha Lal, 1995 and Drisya, and Sathish Kumar, 2018). That said, due to this calibration, some of our transects are lower than what would be expected for some portions of river. However, the average of all transects is 0.025 and the majority of the transects fall between 0.02 – 0.1. We will also note that there is considerable uncertainty in the geometry of the channel. The river bathymetry was last surveyed in 2010, and in this application, merged into a DEM based on Lidar data from 2014. There are therefore other levels of uncertainty likely being absorbed into our calibration of the Manning's coefficient.

To clarify this information, we have added the following text to the revised manuscript's supplemental information, Page 1, Lines 20-25 "In-channel Manning's coefficients are tuned to calibrate the model's resulting water surface elevations with that of the observed water surface data. Manning's coefficients for the rest of the computational domain (e.g., anything overbank) are estimated using 2011 Land Cover data from the Western Washington Land Cover Change Analysis project (NOAA, 2012), and visual inspection of aerial imagery and range from 0.04 (cleared land with tree stumps) - 0.1 (heavy stands of timber/medium to dense brush). These values are extracted from the HEC-RAS Hydraulic Reference Manual, Table 3-1 (Brunner, 2016)" and Page 2, Lines 15-17, "Manning's coefficients within the main channel of the Quillayute River are calibrated to best represent the water surface elevation on the day of the USGS

longitudinal survey. Final Manning's coefficients range from to 0.005 to 0.1, and are on average 0.025."

"How are the high water level events constructed? The possible presence of autocorrelation in the data is not mentioned – it would be good to test for this or report the results of such a test if it has been done already."

Response: As we are not 100% clear on what the reviewer is referring to here our response will focus on how we construct high still water level events.

Still water levels (SWLs) at the downstream boundary are constructed using methods from Serafin and Ruggiero, 2014 and Serafin et al., 2017 but are also described in detail below. The motivation behind our simulations are to generate distributions of many combinations of extreme and non-extreme variables. Based on the modeling techniques used, some signals are simulated with autocorrelation, but most are not. Our focus is on representing the nonstationarities and dependencies in our bulk distributions of the simulated variables (SWLs and all its components, wave height, wave period, wave direction, climate indices, discharge) and determining how combinations of these variables may alter flooding.

SWLs from the tide gauge are first decomposed into mean sea level, tide, and non-tidal residual components. Mean sea level is determined by a linear regression applied to monthly means of the SWL record. Non-tidal residual is comprised of all water level signals not related to the astronomical tide, and is includes the intra-annual seasonal signal, monthly mean sea level anomalies (inter-annual variability), and a high-frequency residual related to storm surge due to atmospheric pressure anomalies and wind setup. The seasonal signal is produced by a regression model that includes annual and/or semiannual harmonics, fit to the SWL time series with mean sea level removed. Monthly mean sea level anomalies are computed once the seasonal signal is removed from the water level signal by averaging each month on record. To extract storm surge after mean sea level, seasonality and monthly mean sea level anomalies

have been removed, two year blocks of the water level time series are transformed into the frequency domain and, following the spectral methods of Bromirski et al., [2003], tide bands are removed and replaced with amplitude and phase estimates consistent with the concurrent nontide continuum. The result is a storm surge time series that excludes tidal and other low frequency energy. The tide was extracted from NOAA's tidal predictions and the annual (Sa) and semiannual (Ssa) harmonic constituents were removed. A SWL time series is then constructed by adding the above decomposed time series back together.

To statistically simulate daily time series of the above components

1) Storm surge is split into extreme (using a peak over threshold approach) and non-extreme components. Extreme storm surge are fit to non-stationary Generalized Pareto Distributions which include seasonality and climate indices as covariates. Non-extreme storm surge are fit to monthly logistic distributions. Storm surge is then statistically simulated using a bivariate logistic model dependent on wave height.

2) Monthly mean sea level anomalies are simulated based on a best-fit, multiple linear regression model to the Multivariate ENSO Index (MEI). Climate indices (e.g., the MEI or the Pacific/North American teleconnection pattern (PNA) which is associated with fluctuations in the jet stream) are simulated using Markov Chains to incorporate auto-correlation into the simulated signal.

3) Daily astronomical tide is simulated from a repeated deterministic tide time series such that we are simulating "modern day" extremes and not including longer term tide cycles in our analysis. The daily maximum tide is selected every day from the repeated time series. The daily maximum TWL occurs during the daily maximum tide approximately 70% of the time, therefore, for 30% of the daily data, a random estimate sampled from an exponential fit to the differences between the daily maximum TWL and the maximum daily tide.

"Other suggestions

Figure 13: the grey dashed lines presumably belong to the 4 different return periods shown – it would be easier for the reader to use the same colours (but dashed) instead of grey."

Response: Excellent suggestion, this figure has been modified.

"Caption of figure 13: "the pink shaded area represents a transition zone, where neither event drives the water level". The last part is not clearly phrased. Do you mean the zone where the water level is not driven by either the coastal or river drivers alone?"

Response: Thanks for catching – the text has been changed to, "The grey shaded area represents a transition zone, where the water level is driven by a combination of SWL and Q events."

"Page 26, lines 14-15: "At low tide, a high river discharge may promote drainage of the floodwater into the ocean (Kumbier et al., 2018), increasing water levels for days at a time and prolonging exposure to flooding". Why would a low tide that promotes drainage to the ocean lead to increased water levels? Would the opposite not lead to backwater effects?"

Response: Thanks for catching this, this statement has been removed from the text and changed to the following Page 21 - 22, Lines 11-12 and 1-2, "The outletting to the ocean as the tide recedes artificially inflating SWLs at the tide gauge, increasing water levels for days at a time and prolonging exposure to flooding. When subtracting a tide time series from this signal, storm surge would appear to be elevated at low tide."

"In the abstract it is stated that "Understanding the relative forcing of extreme water levels along an ocean-to-river gradient will better prepare communities within inlets and estuaries for the compounding impacts of various environmental forcing". A similar statement can be found in the conclusions. I feel that this requires more nuance. There are many steps that would be needed to make these (important) scientific insights usable by a local community for preparing themselves."

Response: We have made this sentence less specific by writing, "Understanding the relative forcing driving extreme water levels along an ocean-to-river gradient will help communities within inlets better understand their risk to the compounding impacts of various environmental forcing, important for increasing their resilience to future flooding events."

"Page 17-line 14-15: "ADCIRC simulations confirm this phenomenon, as the river discharge peak is modeled exactly at low tide (Figure 5)". I find it hard to see that when looking at Figure 5. Maybe help the reader a bit more? For me it seems more to be at high tide but maybe there is something I am missing."

Response: Figure 5 (in the original manuscript) displayed only the storm surge, so lacked tide, mean sea level, seasonality, and monthly sea level anomalies. We have created a second panel within the figure (Figure 7 in the revised manuscript) which also includes tidal level from the ADCIRC simulations to help guide the reader to this conclusion.

Textual changes

"Page 3, line 30. Change ". . .experiencing relative sea level rates of. . ." to ". . .experiencing relative sea level change rates of. . ." (similar comment in line 31)."

Response: This has been corrected.

"Page 8, lines 10-11: add "in most cases"."

Response: This has been corrected.

"Page 8, line 30 (and the rest of the text): where is Toke Point tide gauge on Figure 1?"

Response: These labels were accidentally left off our original Figure. Figure 1 in the revised manuscript now includes labels for all tide gauges, as well as a legend that reflects all mapped features.

"Page 11, line 12. Change "periosd" to "period" "

[Figure]

Response: This has been corrected.

"Page 14, line 13. Change "subsituting" to "substituting""

Response: This has been corrected.

"Page 23, line 19: suggest to remove "regardless of the likelihood" (it is already in the return level events?)"

Response: This has been corrected.

"Page 23-line 5 and 8: add "a" and "b" to Figure 13 to help the reader. References not mentioned in manuscript"

Response: This has been corrected.

References:

Bevacqua, E., Maraun, D., Hobæk Haff, I., Widmann, M. and Vrac, M., 2017. Multivariate statistical modelling of compound events via pair-copula constructions: analysis of floods in Ravenna (Italy). Hydrology and Earth System Sciences, 21(6), pp.2701-2723.

Bromirski, P.D., Flick, R.E. and Cayan, D.R., 2003. Storminess variability along the California coast: 1858–2000. Journal of Climate, 16(6), pp.982-993.

Couasnon, A., Sebastian, A. and Morales-Nápoles, O., 2018. A Copula-Based Bayesian Network for Modeling Compound Flood Hazard from Riverine and Coastal Interactions at the Catchment Scale: An Application to the Houston Ship Channel, Texas. Water, 10(9), p.1190.

Drisya, J. and Sathish Kumar, D., 2018. Automated calibration of a two-dimensional overland flow model by estimating Manning's roughness coefficient using genetic algorithm. Journal of Hydroinformatics, 20(2), pp.440-456.

Leonard, M., Westra, S., Phatak, A., Lambert, M., van den Hurk, B., McInnes, K., Risbey, J., Schuster, S., Jakob, D. and Stafford Smith, M., 2014. A compound event

framework for understanding extreme impacts. Wiley Interdisciplinary Reviews: Climate Change, 5(1), pp.113-128.

Moftakhari, H., Schubert, J.E., AghaKouchak, A., Matthew, R. and Sanders, B.F., 2019. Linking Statistical and Hydrodynamic Modeling for Compound Flood Hazard Assessment in Tidal Channels and Estuaries. Advances in Water Resources.

Wahl, T., Jain, S., Bender, J., Meyers, S.D. and Luther, M.E., 2015. Increasing risk of compound flooding from storm surge and rainfall for major US cities. Nature Climate Change, 5(12), p.1093. Wasantha Lal, A. M. "Calibration of riverbed roughness." Journal of Hydraulic Engineering 121, no. 9 (1995): 664-671.

Zscheischler, J., Westra, S., Hurk, B.J., Seneviratne, S.I., Ward, P.J., Pitman, A., AghaKouchak, A., Bresch, D.N., Leonard, M., Wahl, T. and Zhang, X., 2018. Future climate risk from compound events. Nature Climate Change, p.1.

**Hydraulic Model**
(e.g., HEC-RAS)

**Probabilistic
Simulations of
Boundary Conditions**
(e.g., SR14)

*Water Surface
Elevation*
(at each transect
for a number of
boundary conditions)

*pass synthetic
boundary conditions
into surrogate models*

**Surrogate Models**
(at each transect)

*Probabilistic
Estimates of Water
Surface Elevation*
(at each transect)

**Fig. 1.** Schematic of hybrid statistical-physical modeling technique. Models are portrayed as squares, while circles portray model outputs.

Flow Boundaries

**Legend**

Quillayute Main: 9939.227

**Fig. 2.** Hydrograph for the low discharge unsteady simulation 1. Scenario 1, Low discharge 20% increase in flow, peak flow = 25cms, SWL = -1m

**Fig. 3.** Comparison of the water surface elevation of the maximum discharge in a steady flow run with the water surface elevation during the maximum discharge in an unsteady flow run.

Flow Boundaries

Legend

Quillayute Main: 9939.227

05/01/2019 17:43, 116.70

**Fig. 4.** Hydrograph for the average discharge unsteady simulation 2. Scenario 2, Average discharge 20% increase in flow, peak flow = 120cms, SWL = -1m

**Fig. 5.** Comparison of the water surface elevation of the maximum discharge in a steady flow run with the water surface elevation during the maximum discharge in an unsteady flow run.

River: Quillayute Reach: Main RS: 9939.227

**Legend**
Flow
Q Min + Multiplier

**Fig. 6.** Hydrograph for the extreme discharge unsteady simulation 3. Scenario 3, Extreme discharge 2000% increase in flow, peak flow = 884cms, SWL = -1m

**844cms max**

Fig. 7. Comparison of the water surface profile of the maximum discharge in a steady flow run with the water surface profile during the maximum discharge in an unsteady flow run.

---

## Author Response (AR2)

May 24, 2019

Dr. Paolo Tarolli
Viale dell'Universita, 16
35020 Legnaro PD, Italy

Dear Dr. Tarolli,

This letter accompanies the submission of the REVISED manuscript NHESS-2018-347 entitled, "What's streamflow got to do with it? A probabilistic simulation of the competing oceanographic and fluvial processes driving extreme along-river water levels." We have thoroughly considered and addressed the comments of the reviewers, and feel that this enabled us to better explain and clarify the overall framework and the importance of our results, strengthening the overall manuscript.

Please find reviewer comments given in verbatim and our replies in italics. References to locations in the article are made by section (e.g., S3.5) or line number corresponding to the updated manuscript. The re-revised manuscript, which includes 14 figures, a supplemental information section, and this letter, has been uploaded.

Thank you for your consideration. Please contact me at kserafin@stanford.edu with any questions.

Sincerely,

Katherine A. Serafin (lead and corresponding author)
Postdoctoral Research Fellow
Stanford University

Anonymous Referee #1

General comments

Serafin et al present a new framework for examining the joint influence of several coastal and riverine processes on water levels in estuarine environments, and show very clearly that the 100-yr ocean or 100-yr streamflow event does not always produce the 100-yr along-river water level. It is a novel piece of work using a clever methodological framework, resulting in an analysis that can assesses non-stationary water levels from a multivariate joint distribution and truly decompose coastal water levels. As such, I believe that the research forms an important contribution to the increasingly important field of compound flood risk assessment. The manuscript is well written in terms of language, but parts of it feel to long or could be helped by restructuring. There are also some specific methodological issues that require further explanation, as described in the following review. Nevertheless, if these can be sufficiently responded to, I believe that this paper would provide a very valuable addition to the literature.

*Thank you!*

Main comments

The introduction is generally well written and reviews most of the relevant literature. However, some important concepts for the paper are not fully introduced or defined. For example, a formal definition of compound flooding is missing. On page 2, line 20 (and also later at page 25, line 15) the authors imply that probabilistic simulations of water levels have not yet been done considering ocean and onland processes, and that this has only been done for specific events. However, Bevacqua et al (2017) van den Hurk et al. (2015), and Couasnon et al. (2018) have used probabilistic simulations. The current paper certainly adds value to the research carried out in those studies, but it would be prudent to mention them and how the current study advances.

*We thank the reviewer for pointing out this oversight. We have added a formal definition of a compound event in the first line of the introduction, Page 1, Lines 16-17, "Coincident or compound events are a combination of physical processes in which the individual variables may or may not be extreme, however the result is an extreme event with a significant impact (Zscheischler et al., 2018, Bevacqua et al., 2017, Wahl et al., 2015, Leonard et al., 2014)." We have also added a brief description to the abstract, Page 1, Lines 1 -2, "Extreme water levels generating flooding in estuarine and coastal environments are often driven by compound events, where many individual processes such as waves, storm surge, streamflow, and tides coincide."*

*We thank the reviewer for noting that our writing seemed to imply that we were the first to produce probabilistic simulations of discharge and coastal water level events. Our intent was to highlight the novelty of the hybrid methodology merging physical and statistical models for return level analysis – which at the time the manuscript was completed and submitted was a novel application. However, after submission, Couasnon et al., 2018 and Moftakhari et al., 2019 published complementary frameworks. We thank the reviewer for suggesting the additional references, and have included the following text in our introduction to highlight the variety of previous studies, Page 2, Lines 22 – 28, "On the other hand, statistical models allow*

Department of Geophysics
397 Panama Mall, Mitchell Earth Sciences Building, Stanford, CA 94305  T 650.497.6509

*for the investigation of compound water levels through the simulation of combinations of dependent events which may not have been physically realized in observational records (Bevacqua et al., 2017, van den Herk et al., 2015). In addition, researchers have recently begun to generate hybrid models that link statistical and physical modeling approaches for understanding compound flood events (Moftakhari et al., 2019, Couasnon et al., 2018). Similar to the results solely from hydrodynamic and hydraulic models, statistical and hybrid modeling strategies show that simplifications of the dependence between multiple forcings may lead to an underestimation of flood risk."*

In terms of the overall structure, the methods section (section 3, but also parts of section 4) are sometimes difficult to follow. The really interesting part here is the new overall framework. However, this overall framework sometimes gets lost in the details of the various specific models used, which can be rather lengthy (e.g. the part on HEC-RAS). It would be beneficial to the reader to highlight the overall methodological framework more clearly at the start of the methods section, for example with a flowchart. This would highlight more clearly the major novelty of this paper. It is of course also necessary to give details of the various components used for each part of the framework, such as HEC-RAS. But by emphasizing more the framework, it would be clear that one could also use the overall assessment framework with other hydrodynamic models, if one wished to do so.

Following on, it may help to move some of the details to Supplementary Information. General background information about setting up HEC-RAS can be shortened, and the essential parts for this study could be moved to supplementary information. This would improve overall readability of this section.

Related to the previous comments on structure, the part on HEC-RAS model validation (3.2.1) seems out of place in the methods section. It could be moved to the section on validation or in my opinion better to still to supplementary information.

*We thank the author for the suggestions on how to better emphasize our framework. We have added a flowchart schematic that we hope better explains our hybrid modeling technique in the revised manuscript (Figure 3 in the manuscript, below as Figure 1). We agree that the amount of detail presented in the original manuscript may have added unnecessary length and detracted from the main value of the paper. In the revised manuscript, we have moved the sections describing the HEC-RAS model domain setup, validation and calibration to the Supplemental Information. We also have moved the section describing the tide gauge merging and removal of the river-influenced water levels to the Supplemental Information. We streamlined many of the sections and believe we have improved the overall readability of the manuscript.*

Department of Geophysics
397 Panama Mall, Mitchell Earth Sciences Building, Stanford, CA 94305  T 650.497.6509

[Figure]

*Figure 1: Schematic of hybrid statistical-physical modeling technique. Models are portrayed as squares, while circles portray model outputs.*

My main methodological concern relates to the use of steady flow simulations. As the authors state themselves in the discussion, the steep catchment of the mountainous environment means a short response time for rainfall. It also calls into question the validity of using steady-state flow for the analysis. I would like the authors to explain this choice and explain what it means for the overall results? Has there been any sensitivity assessment of the results compared to an unsteady state simulation, for example?

Department of Geophysics
397 Panama Mall, Mitchell Earth Sciences Building, Stanford, CA 94305  T 650.497.6509

When originally considering merging the statistical and physical model, we started with steady flow scenarios in order to keep our simulations as simple as possible. With millions of combinations of boundary conditions, admitting discharge as a function of time would add another level of complexity to the modeling framework. However, as the reviewer mentions and as we state in the manuscript, the response time can be short, with the river rising to peak flow over 1-2 days.  We completed a sensitivity assessment of steady versus unsteady simulations for a variety of hydrographs and stationary downstream boundary conditions. Figures below compare water surface elevations from the steady flow simulation of the peak discharge condition in each hydrograph to water surface elevations during the peak discharge condition from an unsteady flow simulation. Below we present results from a low, average, and extreme flow scenario (Figures 2, 4, 6, respectively).

Our results show that for many along-river locations, the steady flow approximation is reasonable. Average (standard deviation) differences between water surface elevations are 5 cm (20 cm), 7 cm (11 cm), and 40 cm (76 cm) (Figures 3, 5, 7). Large percent differences between steady and unsteady simulations are due to differences between very small numbers (e.g., Figure 7). These results show that at specific locations, unsteadiness is indeed an area of improvement, but it is currently outside the scope of what has been done. On page 22 and 23, Lines 9-10 and 1-2, we have included the following, "Thus, with co-occurring daily maximum SWL and discharge, our model may miss certain dynamics important for flooding over unsteady conditions. Furthermore, interactions between storm surge and river discharge may increase the overall elevation of the residual (Maskell et al., 2013). While beyond the scope of our present study, these unsteady characteristics are important to consider in future research."

Our methodology at this point is not intended to model any specific event perfectly, but instead to understand locations where compounding SWL and Q are statistically likely to occur together and potentially generate flooding events. This technique can also provide useful information for choosing appropriate boundary conditions for the modeling of unsteady flow scenarios. Finally, as mentioned in the manuscript, the existence of a longitudinal profile of the water surface elevation, which we were able to recreate using steady flow simulations, provided some confidence in our selection of steady flow simulations.

Department of Geophysics
397 Panama Mall, Mitchell Earth Sciences Building, Stanford, CA 94305  T 650.497.6509

[Figure]

*Figure 2: Hydrograph for the low discharge unsteady simulation 1. Scenario 1, Low discharge*
*20% increase in flow, peak flow = 25cms, SWL = -1m*

[Figure]

*Figure 3: Comparison of the water surface elevation of the maximum discharge in a steady flow run with the*
*water surface elevation during the maximum discharge in an unsteady flow run. Large percent change occur*
*when dividing by small numbers.*

**Scenario 2, Average discharge**
**20% increase in flow, peak flow = 120cms, SWL = -1m**

[Figure]

*Figure 4: Hydrograph for the average discharge unsteady simulation 2.*

[Figure]

*Figure 5: Comparison of the water surface elevation of the maximum discharge in a steady flow run with the water surface elevation during the maximum discharge in an unsteady flow run. Large percent change occur when dividing by small numbers*

Department of Geophysics
397 Panama Mall, Mitchell Earth Sciences Building, Stanford, CA 94305  T 650.497.6509

***Scenario 3, Extreme discharge***
***2000% increase in flow, peak flow = 884cms, SWL = -1m***

[Figure]

*Figure 6: Hydrograph for the extreme discharge unsteady simulation 3.*

[Figure]

*Figure 7: Comparison of the water surface profile of the maximum discharge in a steady flow run with the water surface profile during the maximum discharge in an unsteady flow run. The bottom panel is scaled to easily view values; values off the chart are up to 300% different, but for locations where the elevation goes from negative to positive.*

Department of Geophysics
397 Panama Mall, Mitchell Earth Sciences Building, Stanford, CA 94305  T 650.497.6509

It is not clear what Manning's coefficients are used on the floodplains. It is stated that they are estimated using 2011 Land Cover data from the Western Washington Land Cover Change Analysis project (NOAA, 2012) and visual inspection of aerial imagery. But what values were selected for different land use classes? Moreover, on page 8, line 20 the Manning coefficient of "0.005" is very low and not really representative of natural river states. Is there a specific reason for this?

*The Manning's coefficients used over the floodplain ranged from 0.04 (cleared land with tree stumps) - 0.1 (heavy stands of timber/medium to dense brush). These values were extracted from Table 3-1 in Brunner, 2016. For the channel, Manning's coefficients were calibrated on a transect by transect basis to determine the best-fitting longitudinal water surface profile compared to the measured data. This technique of optimizing the Manning's coefficient is widely used in the literature when observational water surface profiles exist (e.g., Wasantha Lal, 1995 and Drisya, and Sathish Kumar, 2018). That said, due to this calibration, some of our transects are lower than what would be expected for some portions of river. However, the average of all transects is 0.025 and the majority of the transects fall between 0.02 – 0.1. We will also note that there is considerable uncertainty in the geometry of the channel. The river bathymetry was last surveyed in 2010, and in this application, merged into a DEM based on Lidar data from 2014. There are therefore other levels of uncertainty likely being absorbed into our calibration of the Manning's coefficient. To clarify this information, we have added the following text to the revised manuscript's supplemental information, Page 1, Lines 20-25 "In-channel Manning's coefficients are tuned to calibrate the model's resulting water surface elevations with that of the observed water surface data. Manning's coefficients for the rest of the computational domain (e.g., anything overbank) are estimated using 2011 Land Cover data from the Western Washington Land Cover Change Analysis project (NOAA, 2012), and visual inspection of aerial imagery and range from 0.04 (cleared land with tree stumps) - 0.1 (heavy stands of timber/medium to dense brush). These values are extracted from the HEC-RAS Hydraulic Reference Manual, Table 3-1 (Brunner, 2016)" and Page 2, Lines 15-17, "Manning's coefficients within the main channel of the Quillayute River are calibrated to best represent the water surface elevation on the day of the USGS longitudinal survey. Final Manning's coefficients range from to 0.005 to 0.1, and are on average 0.025."*

How are the high water level events constructed? The possible presence of autocorrelation in the data is not mentioned – it would be good to test for this or report the results of such a test if it has been done already.

*As we are not 100% clear on what the reviewer is referring to here our response will focus on how we construct high still water level events.*

*Still water levels (SWLs) at the downstream boundary are constructed using methods from Serafin and Ruggiero, 2014 and Serafin et al., 2017 but are also described in detail below. The motivation behind our simulations are to generate distributions of many combinations of extreme and non-extreme variables. Based on the modeling techniques used, some signals are simulated with autocorrelation, but most are not. Our focus is on representing the nonstationarities and dependencies in our bulk distributions of the simulated variables (SWLs and all its components, wave height, wave period, wave direction, climate indices, discharge) and determining how combinations of these variables may alter flooding.*

*SWLs from the tide gauge are first decomposed into mean sea level, tide, and non-tidal residual components. Mean sea level is determined by a linear regression applied to monthly means of the SWL record. Non-tidal residual is comprised of all water level signals not related to the astronomical tide, and is includes the intra-*

Department of Geophysics
397 Panama Mall, Mitchell Earth Sciences Building, Stanford, CA 94305  T 650.497.6509

annual seasonal signal, monthly mean sea level anomalies (inter-annual variability), and a high-frequency residual related to storm surge due to atmospheric pressure anomalies and wind setup. The seasonal signal is produced by a regression model that includes annual and/or semiannual harmonics, fit to the SWL time series with mean sea level removed. Monthly mean sea level anomalies are computed once the seasonal signal is removed from the water level signal by averaging each month on record. To extract storm surge after mean sea level, seasonality and monthly mean sea level anomalies have been removed, two year blocks of the water level time series are transformed into the frequency domain and, following the spectral methods of Bromirski et al., [2003], tide bands are removed and replaced with amplitude and phase estimates consistent with the concurrent nontide continuum. The result is a storm surge time series that excludes tidal and other low frequency energy. The tide was extracted from NOAA's tidal predictions and the annual (Sa) and semiannual (Ssa) harmonic constituents were removed. A SWL time series is then constructed by adding the above decomposed time series back together.

To statistically simulate daily time series of the above components

1) Storm surge is split into extreme (using a peak over threshold approach) and non-extreme components. Extreme storm surge are fit to non-stationary Generalized Pareto Distributions which include seasonality and climate indices as covariates. Non-extreme storm surge are fit to monthly logistic distributions. Storm surge is then statistically simulated using a bivariate logistic model dependent on wave height.

2) Monthly mean sea level anomalies are simulated based on a best-fit, multiple linear regression model to the Multivariate ENSO Index (MEI). Climate indices (e.g., the MEI or the Pacific/North American teleconnection pattern (PNA) which is associated with fluctuations in the jet stream) are simulated using Markov Chains to incorporate auto-correlation into the simulated signal.

3) Daily astronomical tide is simulated from a repeated deterministic tide time series such that we are simulating "modern day" extremes and not including longer term tide cycles in our analysis. The daily maximum tide is selected every day from the repeated time series. The daily maximum TWL occurs during the daily maximum tide approximately 70% of the time, therefore, for 30% of the daily data, a random estimate sampled from an exponential fit to the differences between the daily maximum TWL and the maximum daily tide.

Other suggestions

Figure 13: the grey dashed lines presumably belong to the 4 different return periods shown – it would be easier for the reader to use the same colours (but dashed) instead of grey.

*Excellent suggestion, this figure has been modified.*

Caption of figure 13: "the pink shaded area represents a transition zone, where neither event drives the water level". The last part is not clearly phrased. Do you mean the zone where the water level is not driven by either the coastal or river drivers alone?

*Thanks for catching – the text has been changed to, "The grey shaded area represents a transition zone, where the water level is driven by a combination of SWL and Q events."*

Department of Geophysics
397 Panama Mall, Mitchell Earth Sciences Building, Stanford, CA 94305  T 650.497.6509

Page 26, lines 14-15: "At low tide, a high river discharge may promote drainage of the floodwater into the ocean (Kumbier et al., 2018), increasing water levels for days at a time and prolonging exposure to flooding". Why would a low tide that promotes drainage to the ocean lead to increased water levels? Would the opposite not lead to backwater effects?

*Thanks for catching this, this statement has been removed from the text and changed to the following Page 21 - 22, Lines 11-12 and 1-2, "The outletting to the ocean as the tide recedes would artificially inflate SWLs at the tide gauge, increasing water levels for days at a time and prolonging exposure to flooding. When subtracting a tide time series from this signal, storm surge would appear to be elevated at low tide."*

In the abstract it is stated that "Understanding the relative forcing of extreme water levels along an ocean-to-river gradient will better prepare communities within inlets and estuaries for the compounding impacts of various environmental forcing". A similar statement can be found in the conclusions. I feel that this requires more nuance. There are many steps that would be needed to make these (important) scientific insights usable by a local community for preparing themselves.

*We have made this sentence less specific by writing, "Understanding the relative forcing driving extreme water levels along an ocean-to-river gradient will help communities within inlets better understand their risk to the compounding impacts of various environmental forcing, important for increasing their resilience to future flooding events."*

Page 17-line 14-15: "ADCIRC simulations confirm this phenomenon, as the river discharge peak is modeled exactly at low tide (Figure 5)". I find it hard to see that when looking at Figure 5. Maybe help the reader a bit more? For me it seems more to be at high tide but maybe there is something I am missing.
*Figure 5 (in the original manuscript) displayed only the storm surge, so lacked tide, mean sea level, seasonality, and monthly sea level anomalies. We have created a second panel within the figure (Figure 7 in the revised manuscript) which also includes tidal level from the ADCIRC simulations to help guide the reader to this conclusion.*

Textual changes

Page 3, line 30. Change ". . .experiencing relative sea level rates of. . ." to ". . .experiencing relative sea level change rates of. . ." (similar comment in line 31).

*This has been corrected.*

Page 8, lines 10-11: add "in most cases".

*This has been corrected.*

Page 8, line 30 (and the rest of the text): where is Toke Point tide gauge on Figure 1?

*These labels were accidently left off our original Figure. Figure 1 in the revised manuscript now includes labels for all tide gauges, as well as a legend that reflects all mapped features.*

Page 11, line 12. Change "periosd" to "period"

*This has been corrected.*

Page 14, line 13. Change "subsituting" to "substituting"

*This has been corrected.*

Page 23, line 19: suggest to remove "regardless of the likelihood" (it is already in the return level events?)

*This has been corrected.*

Page 23-line 5 and 8: add "a" and "b" to Figure 13 to help the reader. References not mentioned in manuscript

*This has been corrected.*

Bevacqua E, Maraun D, Haff I H,WidmannM and Vrac M 2017 Multivariate statistical modelling of compound events via pair-copula constructions: analysis of floods in Ravenna (Italy) Hydrol. Earth Syst. Sci. 21 2701-2723.
Couasnon A, Sebastian A and Morales-Nápoles O 2018 A Copula-based bayesian network for modeling compound flood hazard from riverine and coastal interactions at the catchment scale: An application to the houston ship channel, Texas. Water, 10, 9, 1190
 Van den Hurk B, van Meijgaard E, de Valk P, van Heeringen J and Gooijer ´J 2015 Analysis of a compounding surge and precipitaiton event in the Netherlands Environ. Res. Lett. 10, 035001

*Above suggested references have been included in the text.*

Anonymous Referee #2

The paper overall presents a good contribution, however, it needs some work Some concepts are not clear, and the reader is left 'guessing' about their meaning. for example, in the introduction, a reader is not aware of what 'bivariate or multivariate processes' are, thus they can't understand the challenge in trying to identify them or study them.

*We thank the reviewer for pointing out the lack of contextual information in the initial submission. Bivariate and multivariate processes are processes that occur from two or multiple variables, respectively. In coastal environments, multiple processes like waves, tides, storm surge, and river discharge, may combine to drive an extreme flood event. We have improved the clarity of our descriptions of multivariate and bivariate processes by removing the sentence driving confusion (Page 1, Line 21-22 original manuscript) while introducing a formal definition of a compound event in the first line of the introduction, Page 1, Lines 16-21, "Coincident*

Department of Geophysics
397 Panama Mall, Mitchell Earth Sciences Building, Stanford, CA 94305  T 650.497.6509

*or compound events are a combination of physical processes in which the individual variables may or may not be extreme, however the result is an extreme event with a significant impact (Zscheischler et al., 2018, Bevacqua et al., 2017, Wahl et al., 2015, Leonard et al., 2014). Flooding is often caused by compound events, where multiple factors impact both open coast and estuarine environments. Storm events, for example, often generate concurrently large waves, heavy precipitation driving increased streamflow, and high storm surges, making the relative contribution of the actual drivers of extreme water levels difficult to interpret." We have also added a brief description to the abstract, Page 1, Lines 1 -2, "Extreme water levels generating flooding in estuarine and coastal environments are often driven by compound events, where many individual processes such as waves, storm surge, streamflow, and tides coincide." We hope that this revision will help readers to understand the types of events we are focused on understanding.*

My major concerns are related to the method section, that currently needs much improvement. In its present state, it is much too long in some parts, and not enough clear on the overall framework, which is the added value of this work. There is far too much description of known elements, such as HEC-Ras, for example, and not enough clarity on the proposed approach. Also, it is not too clear if chapter 4 is a method or a discussion of results. As a consequence, it is very hard to understand the discussion of the results.

*We agree that the amount of detail presented in the original manuscript may have added unnecessary length and detracted from the main value of the paper and point out that Reviewer 1 had a very similar comment. Therefore, in the revised manuscript, we have moved the sections describing the HEC-RAS model domain setup, validation and calibration to the Supplemental Information. We also have moved the section describing the tide gauge merging and removal of the river-influenced water levels to the Supplemental Information. Section 4 in the original manuscript was difficult to interpret, so we merged the text from this section in with methods, results, and discussion sections in the revised manuscript in a fluid way. We have also added a schematic of the hybrid-modeling framework (Figure 3, revised manuscript and below), to help to clarify and emphasize the overall framework for readers.*

[Figure]

*Figure 3: Schematic of hybrid physical-statistical modeling technique. Models are portrayed as squares, while circles portray model outputs.*

Department of Geophysics
397 Panama Mall, Mitchell Earth Sciences Building, Stanford, CA 94305  T 650.497.6509

domain boundary conditions are chosen as the water surface elevation at the tide gauge (m; downstream boundary) and river discharge from a combination of records representing the Quillayute River watershed (m$^3$s$^{-1}$; upstream boundary).

**1.1 HEC-RAS model validation**

In order to determine the dominant inputs to Quillayute River discharge, combined estimates of the Sol Duc and Calawah Rivers are compared to measurements taken on the Quillayute River in May 2010 (Czuba et al., 2010). Combined discharge estimates from the Sol Duc and Calawah rivers underpredict streamflow in the Quillayute River by approximately 33%. An area scaling watershed analysis (Gianfagna et al., 2015), described in the main text, found that the Bogachiel and Calawah Rivers had similar contributions. Thus the Calawah river is scaled by a factor of 2.09 to represent the Bogachiel River. Combined discharge estimates from the Sol Duc River and Bogachiel River, representing the Quillayute River, are also compared to the Quillayute discharge measurements taken during the 2010 survey. Using this methodology, the discharge estimates of the Quillayute River fall within the uncertainty of the discrete USGS measurements in most cases (Table 1).

[revised manuscript text omitted]